# Investigating the Influences of Smart Port Practices and Technology Employment on Port Sustainable Performance: The Egypt Case

**Alaa Othman** [1,2,*] **, Sara El Gazzar** [1] **and Matjaz Knez** [2]

1 College of International Transport and Logistics, Arab Academy for Science, Technology and Maritime Transport, Alexandria 1029, Egypt
2 Faculty of Logistics, University of Maribor, 3000 Celje, Slovenia
* Correspondence: alaa.abomousa@gmail.com

**Abstract:** In this digital age, sustainable development and competitive advantage of port businesses rely on their capacity to adapt to changing business requirements. Although many previous studies develop a relation between smart port, technology, and sustainable performance, there is an urgent need to address such issues practically among ports, especially in developing countries. Therefore, this research aims at investigating to what extent the Egyptian ports could apply the smart practices and employ technology to achieve and improve sustainable port performance. The Egyptian ports have been selected to serve as an empirical study in this research, assessing their current performance and evaluating their level of readiness and adaptability to smart practices and technology employment. Interviews have been conducted with a group of 10 different stakeholders from government, private sectors, and experts in the field of port management. The interview results showed the main challenges and obstacles that might face the adaptation of technology and sustainable practices in Egyptian ports. This was followed by a focus group with experts in the field to discuss and conclude some procedures that can be adapted to facilitate the implementation of smart practices and technology employment in the Egyptian ports in order to improve their sustainable performance from different perspectives (economic, social, and environmental), deal with the obstacles facing adaptation, and suggest solutions The research adopted 'content analysis' for both phases. The research showed the great potential of technology employment to achieve sustainable performance in Egyptian ports while highlighting the main obstacles and challenges that might face the adaptation with suggestions and recommendations to those obstacles in order to adopt the digital transformation towards a sustainable smart port performance. Nevertheless, there are some limitations that could be an open issue to future researchers and practitioners who can benefit from those suggestions to employ technology and adapt sustainable procedures in ports, foster new practical research initiatives to adapt the smart port practices in different countries, and test their impact on port sustainable performance.

**Keywords:** smart port practices; sustainable performance; technology; Egyptian ports

## 1. Introduction

Sustainability in transportation usually contributes to sustainable development of a community. Ordinarily, the transportation infrastructure development is based on minimizing the initial operation costs and asserting traffic mobility, considering social and environmental requirements [1]. Many efforts have been made to connect the concept of sustainable development to transportation, concentrating on the elements of sustainability in transport [2].

Maritime ports have faced many challenges since a port operates in the middle of a complicated network of interrelated transportation, industrial, and civil infrastructure as a regional multimodal intersection of word-wide supply chains. Consequentially, it faces

multidimensional difficulties to provide efficient, cost-effective, and sustainable means of transporting goods internationally [3].

The pandemic of COVID-19 has affected the overall maritime performance in the upstream and downstream levels since the maritime industry plays a vital role within the short-run emergency response to the pandemic in sustaining positions, international commerce, the international economy, and elaborate sustainability and resilience of ports and maritime transport during and after the pandemic. The outbreak of COVID-19 has impacted the operations, shipping crew, cash flow, production, and delivery in international transport and the shipping industry in direct and indirect ways. COVID-19 has made the world realize how strongly we rely on the interactions among humans to finish work. Businesses that are demanding labor, such as retailing, warehousing, manufacturing, and logistics, are the worst influenced. Therefore, principal technologies of the Fourth Industrial Revolution have become very important [4].

The concept "Industry 4.0" (4IR) rely on the establishment of cyber-physical production systems, where the compatibility among systems, people, and environment should enable real-time transaction ability and decision making [5]. The occupancy of the 4IR is unlimited, controlled by the full-force emergence of technologies. Using 4IR, the future of seaports is experiencing bold transformations; the competitive industry of shipping and port logistics are following several efforts to attain competitive advantage through the 4th IR and expand into a new business area [6].

Ports are the backbone of the country's foreign trade and its gates to the globe. They are considered as the main connection in the multimodal transport chain, in addition to their essential role in extending the economic development process. Strategic aspiration strategies have been made for the growth of seaports and the transformation into a multinational logistics center; this contributes to delivering attractiveness for investment that improves the position of any country to have its ports reach the ranks of the world's advanced ports–categorized as green ports, and to develop a new generation of ports–the generation of Smart Ports [7].

During the past few decades, smart transport systems have achieved a great interest in making transport systems safer, cleaner, more efficient, and innovative. Smart transport systems entail the technologies and strategies that execute the services, by technically using a mix of technologies to monitor traffic conditions, connect with vehicles and centers, and efficiently handle and maintain traffic operations [8].

Major transport enterprises are heavily investing in smart technologies that are considered as the enablers for the digital transformation in the setting of Industry and Logistics 4.0. Large ports in the European situation are already aware of digital databased technologies such as Block chain or Internet of Things (IoT). It is also essential for small and medium-sized ports to take the opportunity to involve these technological solutions in order to combine themselves in a sustainable way into global supply chains [9]. By applying these technologies that converted document management and decision-making procedures to a fully electronic format, the maritime industry has been significantly enhanced in terms of efficiency and reliability of trade and transport, which has improved its overall sustainability performance [8].

Along with sustainability, digitalization is also becoming of the utmost importance for the sea transport sector. Digital activities have been introduced via the Industry 4.0 concept. The Industrial Revolution 4.0 is a permanent economic transformation which means that the usual technological limits are being deleted. It is relying on combining the physical infrastructure with software, sensors, nanotechnology, or digital intelligence technology. Transformation of the port's digitalization increases the connection between the logistics chain, raises the automation of port operations, and facilitates emergencies [10]. In addition, it increases the ability and quality of decision-making, which is based on data, through the use of smart and collaborative platforms [11].

The smart port is a broad concept that contains several aspects of port activities and has different integration levels. Particularly, when it comes to the novel innovative idea of smart

port emergence, which presently has obtained a growing attention in previous studies and research, the investigation of digitalization and interconnected novel technologies becomes essential. The smart port development idea is related to an innovative initiative. Thus, it is important to develop a comprehensive smart index, apply smart practices and employ technology to objectively evaluate its performance, achieve and improve port sustainable performance, obtain advancement opportunities, and completely connect and integrate with the port environment (i.e., all stakeholders of the industry), as well as other ports and logistics performers around the world. Consequently, without the inclusion of small and medium-sized ports, this innovative idea will stay unachievable [9].

Despite the fact that the smart port concept has attracted widespread interest from both industry and academia with a growing number of experimental use-cases emerging from developed countries, few researchers have developed the conceptual framework and indicators of the smart port [12]. Ports in developing countries should consider changes in global trade. In addition, they need to invest in human, institutional, and technological dimensions [13]. Otherwise, this will result in losing their competitive advantages. Smart city and port investigations have become a topic for the research fields in developing countries and Arabian countries. Developing countries usually pursue useful tools for urban growth to face infrastructure inefficiencies, continuous environmental shortages, and insufficient governance tools. Consequently, involving new smart technologies is essential for all small and medium-sized ports in developing countries, and contributions are needed to identify smart port challenges and obstacles for adaptation.

Egypt is one of the emerging economies in the developing countries that has gone through transitional stages, however it faces many challenges concerning transformation [14]. Therefore, this present study aims to investigate to what extent the Egyptian ports could apply smart practices and employ technology to achieve and improve port sustainable performance through conducting an empirical study on Egypt. The rest of the paper is organized as follows: In the next section, a literature review on smart ports, technology employment, and sustainability performance will be provided with a focus on the most comprehensive smart port index and performance measures developed by previous studies, followed by an illustration of the current situation of the Egyptian ports, upon which SWOT analysis will be developed and suggestions to employ technology and smart practices in the Egyptian port will be provided. The paper will conclude with the suggestions and recommendations of some procedures to be conducted for the adaptation of digital transformation towards a sustainable smart port performance, and finally, the last section will address the limitations of this study and suggestions for future studies.

## 2. Literature Review

### 2.1. Smart Port and Sustainabile Performance

Sustainable transport improves the environment, society, and the economic dimension through enhancing green, efficient, and competitive transportation in an integrated way to assure synergies, complementarities, and coherence [15].

Transportation and economic aspects cooperate easily with each other; the transport sector is a vital element of the economy, directly affecting development and the prosperity of populations. Those two members cannot be disunited from each other. Efficient transportation reduces costs and time, increases safety, enhances the economic progress, and allows companies to access the markets in a better way, while inefficient transportation increases costs and decreases company's opportunities (considering countries' macroeconomics as an investment in the transportation segment plays an important role in increasing the GDP and value added).

Transportation also carries an important social and environmental load that cannot be ignored, since the transportation system is primarily there to satisfy human needs without discrimination. Sustainable transportation can help in reducing the median-income community as the expansion of infrastructure creates new employment opportunities [1].

The UN General Assembly adopted the 2030 agenda for sustainable development containing its 17 Sustainable Development Goals (SDGs) with 169 targets, which are "integrated and indivisible, global and universally applicable". Sustainable and resilient transport has arisen in many of the goals and targets; on the one hand, SDG 9 adopts the idea of building resilient infrastructure, promoting inclusive and sustainable industrialization, and fostering innovation; on the other hand, SDG 13 promotes the idea of taking urgent actions to combat climate change and its consequences; another goal is SDG 14 that fosters the conserving and sustainably using of oceans, seas, and marine resources waste [16].

Ports face many challenges in different sectors, including operations (e.g., congestion, delays, operating mistakes, and insufficiency of information sharing), environment (e.g., air, water, and noise pollution, waste removal, building and expansion activities), energy (e.g., heavily energy consumption and energy costs), safety (e.g., berthing effects and vessel crashes), security (e.g., armed thievery, cyber security problems, unlawful actions, and terrorist attacks), and human resources (e.g., lack of awareness and low educational level). As a result of the current problems, ports are adopting technology-based solutions as well as new methods in port operations planning and management [17].

Cyber shipping establishes the evolution of transforming traditional shipping business toward the digitized industrial paradigm considered by industry 4.0. To create value from digital information, hence, it is necessary for traditional shipping to apply new technologies and novel methodological strategies to maintain their infrastructures, workforce, and operations, with a substantial change in their corporate culture and decision-making processes [5].

Digital transformation of ports is not protected by solely utilizing innovative technologies. It is more the cooperation of management benchmarks and workers' knowledge and skills, in addition to applicable IT approaches and techniques, that guarantees sustainable development in order to achieve a smart port [9].

Aslam, Michaelides [18] indicates that a port is identified as "smart" if all the port objects are adequately connected via the Internet; wireless devices, smart sensors, actuators, data centers, and other IoT-based port devices are considered as the main infrastructure of establishing smart ports; in addition, using smart techniques and applications will enhance the overall performance of the port, such as IoT in smart warehouses; RFID sensors that can track and trace reveal location information and automatic allocation for goods in the storage space.

Chuang and Huang [19] studied employing green information technology capital on the relationship between environmental corporate social responsibility and environmental performance. The analysis of the data supported the employment of green IT structural capital, and green IT relational capital on the relationship between environmental corporate social responsibility and environmental performance.

Yau, Peng [20] investigated the influence of smart port infrastructure in improving sustainable performance. Smart ports, as high-performing ports, use information and communications technology [21] to provide a wide range of smart applications, resulting in dramatically enhanced vessel and container management, among other things, and therefore improving the national economy's competitiveness and sustainability. The application of the Internet of Things, cold ironing, renewable energy generation and storage, energy management, container tracking systems and code recognition, the use of AIS data for trajectory forecasting, and resource management are all recent studies on smart ports. Therefore, in order to increase sustainability, ports needs to implement technology-based solutions as well as novel methods to port operations planning and administration [22].

Moosa and He [23] aimed to study the mediation role of environmental technology on the relationship between green operation and sustainable quality performance. The hypotheses were tested using a survey-based technique using a structured questionnaire with closed-ended questions in the study. After the analysis, the result supported the mediation role of environmental technology on the relationship between green operation and sustainable quality performance. In addition, Li and Hu [24] focused on the technological

innovation mediating the relationship between environmental regulation and economic development, where the findings of the study supported that technological innovation mediates the relationship between environmental regulation and economic development.

Moreover, Kumar and Bhatia [25] investigated the mediating role of organizational and technological factors in the relationship between environmental dynamism and performance. The analysis process indicated that the mediating role of organizational and technological factors in the relationship between environmental dynamism and performance was supported.

Based on the above discussion, it has been proved from previous studies that the technology variable employs and effects the relationship between smart port practices and port sustainable performance, which enhance and improve the overall port performance; it is concluded that the technology variable mediates the relationship between the smart port index and port sustainable performance. Different approaches, models, and measures have been proposed by many researchers to employ technology as a mediator to enhance port sustainable performance, which will be illustrated in next subsection.

### 2.2. Smart Port Practices and Performance

Smart technologies have become a prevailing model of information technology and have been utilized to the port industry with the beginning of the Fourth Industrial Revolution [7]. Smart ports describe a port where automatic container terminals utilize smart sensing techniques to enhance performance which are related to different terminal tasks [26]. Molavi [26] and Lim [27] proposed a concept of smart ports that includes an assortment of high-level digital technologies, including monitoring, control, automation, and intelligent equipment, to enhance the port operations and re-energize the current infrastructure for a strengthened port through identifying specific evaluation indicators for the smart port in four main areas–namely operations, environment, energy, and safety and security–in order for ports to develop their smart port strategies and identify the strengths and weaknesses of their current operations in order to achieve continuous improvement.

The study of Rodrigo González, González-Cancelas [22] developed indicators that allow for the measuring and ranking of Spanish smart ports. The index of smart ports related to four pillars: operational economic; social; political and institutional; and environmental. In addition, the study focused on digitalization and new information and communication technologies. Modern remote sensing technologies such as RFID used for identification and localization, cameras, and computer vision algorithms increase safety and reduce operation handling time [26].

The study of Karli and ÇELİKYAY [28] confirmed the relationship between smart seaport dimensions (operation, environment, energy, finance, and safety and security) and sustainability, while the process in ports has been hastened by digitization in many domains, such as Industry 4.0 and smart cities. The study also highlighted that the focus on sustainability distinguishes the smart port transformation from a basic technological shift. Safety and security can be enhanced by employing many possible solutions for structural health structure monitoring; further, Robust wireless networking and Internet connectivity present different communication solution [26].

The study of [20] presents a review of the application of information and communication technology in smart ports. The improvement of ICT can be investigated and used to enhance smart port activities and services, including smart vessel, container, and port management through reducing greenhouse gases emission to rise port sustainable performance. The study revealed that using information systems and technology, including information-gathering application, data centers, networking and communication, and automation, is an essential part of the smart port.

Concerning the above-mentioned previous studies, there is a relationship between smart ports practices, technology, and sustainability, and after counting the most important research that captures the idea, it was found that one of the most comprehensive studies is the study of [7]. This review was dedicated to the majority of the previous literature;

it proposes a theoretical framework of an integrated smart port index linking to port sustainable performance. The review focused on the relationship between smart seaport practices (operation, environment, energy, safety and security, and human resources) and port sustainability performance with its dimensions (economic, social, and environment); the study reveals that it is necessary to adopt an integrated smart port index linking to port sustainable performance, to facilitate port activities and enhance port service possibilities. It is also highlighted that there have been many previous studies on the subject in the past years, especially in developed countries, while it has been regarded that few studies have been made in developing countries. However, one of its limitations is that the index did not include technology employment and its influences on the relationship between smart port practices and port sustainable performance. On the other hand, the study theoretically tests the relationship among variables, despite the researcher suggesting that there is an urgent need to test the proposed theoretical framework of the integrated smart port index and that empirical research needs to be extended to test the applicability of the model presented.

The importance of the smart ports in developing countries has been illuminated by [29]. In their research, the authors demonstrated that current scientific literature in developing countries mainly focuses on manufacturing industries, with little attention paid to the selection technology for smart port in developing countries [29]. Additionally, Philipp [9] demonstrated that small–medium ports have no or limited knowledge on what 4.0 IR smart port practices are and what possibilities they may bring.

Based on the above discussion, it has been found that there is still a need for an integrated performance measurement system capturing the relationship between smart port and sustainable performance while employing technology. Previous studies have proposed different approaches, models, and measures; however there has previously been a limited practical vision that fails to grasp the full picture of technology employment, integrated smart practices, and port sustainable performance. Although paper [7] developed an integrated smart port index capturing different elements of smart port and linking them to port sustainable performance from a different perspective, it was still limited regarding theoretical insight; thus, there is a need to employ technology as a mediating factor as well as to practically address the applicability factors instead of the theoretical factors.

Consequently, it is worth further investigation to capture the integrated relationship from a practical perspective. Therefore, this paper will follow the smart port practices proposed in the above mentioned study [7] as the drivers to sustainable port performance through conducting an empirical study on Egyptian ports.

### 2.3. Egyptian Ports

Smart city and port investigations have become a topic for the research fields in developing countries and Arabian countries. Developing countries usually pursue useful tools for urban growth to face infrastructure inefficiencies, continuous environmental shortages, and insufficient governance tools. These lead to the need to understand the local urban development issues to set a suitable model. It is better to apply a region's specific developmental characteristics when performing its urban renovation via the transfer of knowledge and know-how, rather than simply applying previously operated models used in developed countries [14].

Maritime transport and associated logistics services play an essential role in Egypt's economy and international trade. Almost 90% of the country's international seaborne trade volume takes place through maritime transport. The development in the maritime transport sector is being supported by the growth in major industries such as oil and natural gas, textiles, food processing, and construction [30].

Since 2000, the Egyptian government has set policies targeting the enhancement of the Egyptian exports and captivating neighboring and international markets for container handling, logistics operations, and transshipment [31].

Egypt is considered an essential role-player in the international energy market; by utilizing the Suez Canal and the Suez Mediterranean Pipeline (SUMED), it is a major transit

route [32]. Based on the Integrated Sustainable Energy Strategy ISES 2035, renewable energy capability should contribute 42% of power capacity by 2035. Renewable energy has a major role to play, which is described in the (ISES) to 2035, issued by the Ministry of Electricity and Renewable Energy in 2015.

Egypt Vision 2030 expresses a starting point for the way towards comprehensive development, thus fostering a prosperous pathway through economic, social, and environmental dimensions; Egypt's sustainable development strategy (SDS) represents a roadmap for increasing competitive advantage to reach sustainability in the maritime sector; by 2030, the programs and projects for economic development will include transforming Egypt into an international digital hub, developing the maritime transportation sector, boosting innovation in the energy sector, and adopting an inclusive program to foster innovation and knowledge culture.

In the social dimension, development will include extending the function of state authorities related to transparency and protection, developing communication and technological infrastructure to enhance health care systems, improving the human resource management system, and improving the quality of education and training skills.

In addition, from the environmental perspective, it will include extending the infrastructure for supporting a sustainable water system, increasing the awareness to protect the environment and natural resources, providing incentives for better advanced resources and technologies for water protection, improving the efficiency of the solid-waste management system and supporting its sustainability, creating a system for disposal of hazardous wastes, enhancing the efficiency of saving coastal and marine areas, and adopting policies to decrease air pollution according to climate change [33].

Egypt has entered the 20 20 20 agreement whose goals are a 20% growth in energy efficiency, 20% decrease of $CO_2$ emissions, and 20% renewables by 2020. It relies on the reconfiguration of the electricity grid into a "smart grid" [34].

Egypt contains 82 seaports divided ascendingly according to their scale into main seaports and commercial ports, which are Alexandria, Dekhela, Safaga, East Port Said, Damietta, Adabiyya, Suez, Ain El-Sukhna, Arish, Tur, Hurghada, Sharm Elshiekh, Nuweiba, and other 76 specialty seaports that acquire fishing vessels, oil tankers, mining materials carriers, pleasure boats, tourist yachts, or other ports of a special nature [35].

Egypt has 15 commercial ports that have the total berths' length of main maritime commercial ports of 32.4 km [36]; the total area of major maritime commercial ports is 481.54 km$^2$: 27 specialized ports; 7 mining ports; 4 fishing ports; 11 petroleum ports; and 5 tourist ports. These ports are distinguished by their strategic geographical locations and the approachability of promising investment projects in the coming years, and, consequently, the increases in their competitive advantage [37].

Egypt has launched an automated system for the first time to trace goods till the final release phase and to deal in customs with cargoes contracted with an up-to-date e-commerce system. This system has the opportunity of pre-clearance and charge of customs duties prior to the arrival of the goods.

The Ministry of Communication and Information Technology (MCIT) has an initiative named "Our Future is Digital" and seeks to qualify 100,000 young Egyptians and to expand their ICT skills. Furthermore, Misr Technology Services "MTS" launched the National Single Window for Foreign Trade platform (Nafeza) under the authorization of the Ministry of Finance of Egypt in 2020. The MTS is the layout the ACI documentation procedure, and the CargoX Platform for Blockchain Document Transfer (BDT); it is a national platform that covers Egyptian seaports, land ports, dry ports, and free zones following international standards and best practices.

The National Single Window for Foreign Trade Facilitation "NAFEZA" is an integrated information platform. It combines and coordinates systems and information exchange among all parties included in Egypt's foreign trade system. It authorizes the business and the trade society to submit all documents and transactions for customs, control authorities,

and ports via an online portal to satisfy all regulatory requirements concerned with the discharge of goods [38].

Egypt is one of the emerging economies in the MENA region and has the potential for a transformation path. The country has started many plans and projects for all sectors beginning in 2013, attended by its political and economic changes and urban and regional growth as the top of these priorities. Developing countries faced many obstacles in the last three decades, from population growth, rural pressures on the cities areas, and the fragile economic situation for some cities [14].

Ports began to pay attention to smart solutions to improve operations, enhance efficiency and sustainability, and avoid safety and security incidents. Ports began to turn to the presence of technological solutions to solve the problems faced by the ports, and the transition to smart ports was made [20]. Based on the previous review of the most recurring themes in the literature, it is obvious that choosing a development policy for the Egyptian case as one of the developing countries with transitional stages is necessary due to the following reasons:

- Egypt is considered as one of the emerging economies in developing countries.
- Egypt's maritime industry plays a pivotal role in the economic development of the country.
- Its geographical location is of great importance at the borders of three continents, Europe, Asia, and Africa, linking these by the Nile River and the Suez Canal.
- Egypt has more than 45 seaports, 15 of which are used for commercial objectives.
- Egypt also has 44 additional ports that foster vital economical industries including the fishing sector, mining, petroleum, and tourism.
- Egypt suffers from many challenges that face the improvement of the country's maritime transport.

It has been illustrated that the extent to which smart practices and digitalization can be applied, adopted, and sustained needs to be investigated in the Egyptian context. However, the current status that determines the implementation of smart practices in the Egyptian ports system remains insufficient due to limited academic research and literature that addresses the current situation of the Egyptian ports' performance and identifies the main challenges for smart practices' adaptation from different perspectives; attention should be paid to the truth that various ports may encounter difficulties in implementing a proper digitalization level, which may be affected by several determinants including operations, environment, energy, safety and security, and human resources.

There is a severe lack of investigations and analyses as to the failure obstacles and problems, analyze these causes, and critically suggest solutions to them; therefore, this research is trying to fill this gap while testing, at the same time, the applicability factors instead of theoretical factors, by investigating to what extent the Egyptian ports could apply the smart practices and employ technology to achieve and improve port sustainable performance, through conducting an empirical study on the Egyptian ports. The research will investigates the level of readiness and adaptability through the five dimensions that were previously measured and analyzed in [7]–one of the most comprehensive studies dedicated to the majority of the previous literature–and tackle and measure these variables in detail; the research has adopted this study as a guideline. The results will highlight the obstacles and suggest solutions and recommendations that could enhance the ports' efficiency and their competitive position as well as increase the opportunity for adaptation.

## 3. Materials and Methods

This study is conducted to investigate to what extent there is a level of readiness of adaptability of smart port practices and technology employment to improve port sustainable performance in the Egyptian context. Through conducting an empirical study on Egypt, the research aims at answering the following research questions:

- RQ1: To what extent is there a level of readiness of adaptability of smart port practices and technology in the Egyptian ports?

- RQ2: What are the main obstacles of adapting smart port practices and technology in the Egyptian ports?
- RQ3: What are the procedures and suggested actions that might facilitate the adaptation of smart port practices and technology in the Egyptian ports?

Answering these questions contributes to enhancing the understanding of the Egyptian ports' performance. Accordingly, the Egyptian government and policymakers can develop and design policies and action plans to promote smart practices and technology based on this analysis and understanding. In addition, this study contributes to expanding the research contributions regarding the practical side of smart ports in developing countries, particularly in the Middle East.

### 3.1. Research Design

Smart port adaptation has not been sufficiently investigated in the scientific literature until now, and the concept is yet to receive the attention needed regarding ports in developing countries. Accordingly, to the best of our knowledge, following Gregar [39], if a concept or phenomenon needs to be understood because little research has been conducted on a specific topic, the qualitative research method is significantly useful. Thus, in the path of the current study, the decision was made to select a qualitative research approach. Hence, qualitative research in this research represents narrative research in addition to secondary data from previous research and international reports, reviewed in the previous section

An empirical study was conducted to investigate to what extent the Egyptian ports could apply the smart practices and employ technology to achieve and improve port sustainable performance. Primary data were collected from two research phases based on data collected, analyzed, and evaluated stages; the first phase, which is a semi-structured interview, analyzed the current situation, evaluated the level of readiness and adaptability to smart practices and technology employment in the Egyptian ports' context, and investigated the obstacles and procedures; then the second phase, which is a focus group, discussed and concluded some procedures that can be adapted to facilitate the implementation of smart practices and technology employment in the Egyptian ports and identified the obstacles facing adaptation to draw suggestions and recommendations.

In the frame of narrative research, the collected information was concentrated and summarized. Building upon this, the research was completed by conducting semi-structured interviews and focus group discussions to converse with experts and collect elicited data. Interviews are the most commonly used data collection method in qualitative research and the foremost method in narrative investigation in particular, as the interviewees answer the questions relying on their narrative schema that reflects their knowledge and experiences; in addition, based on [40], focus group discussion was used to explain and expand findings. The semi-structured interviews and the focused group were analyzed using content data analysis methods, in addition to a comprehensive literature review of related theories and approaches, topic-related policy regulations, and guidelines.

Figure 1 summarizes the framework for the research methodology of the study and its relationship to the research questions, while the results of the analysis of the primary data and answering the research questions are reviewed in the next section.

Concerning the methodological strategy, one of the most comprehensive studies is the study proposed by [7], which is dedicated to the majority of the previous literature, resulting in proposing a theoretical framework of an integrated smart port index linking to port sustainable performance. The study revealed that five main groups govern the smart port–the environment, operation, safety and security, human factor, and energy– and confirmed that by adopting the smart port index, safe, economical, and sustainable improvement will be enabled. This ensures that it fulfills the needs of recent and future generations in terms of economic, social, and environmental dimensions.

On one hand, the five domains of smart port practices are considered as independent pillars, while the sustainable port performance is considered as a dependent pillar. On the other hand, as mentioned above, technology is considered to be a mediating role between

smart practices and port sustainable performance. Consequently, the scheme of the research framework is illustrated in Figure 2

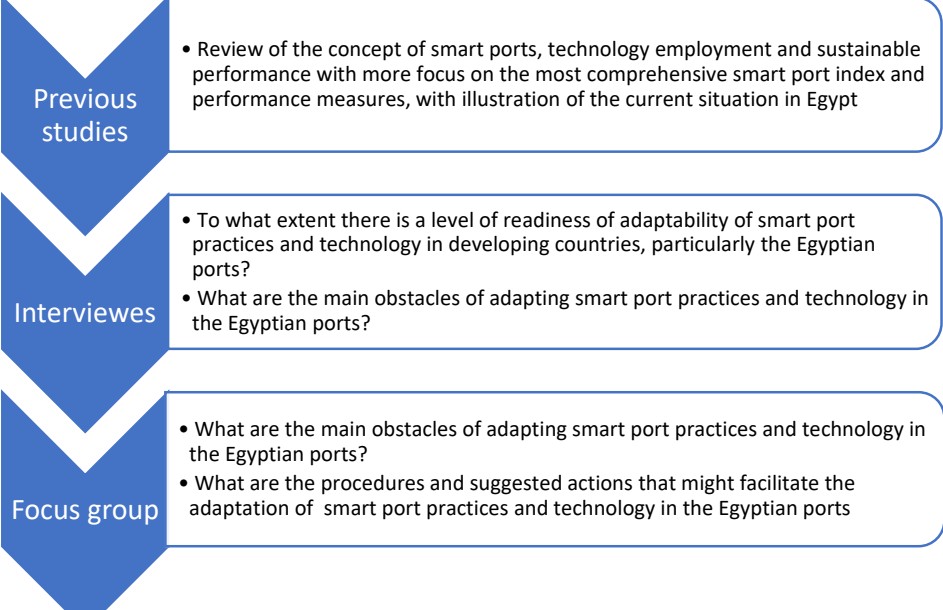

**Figure 1.** The research methodology of the study and its relationship to research questions. Source: The researchers.

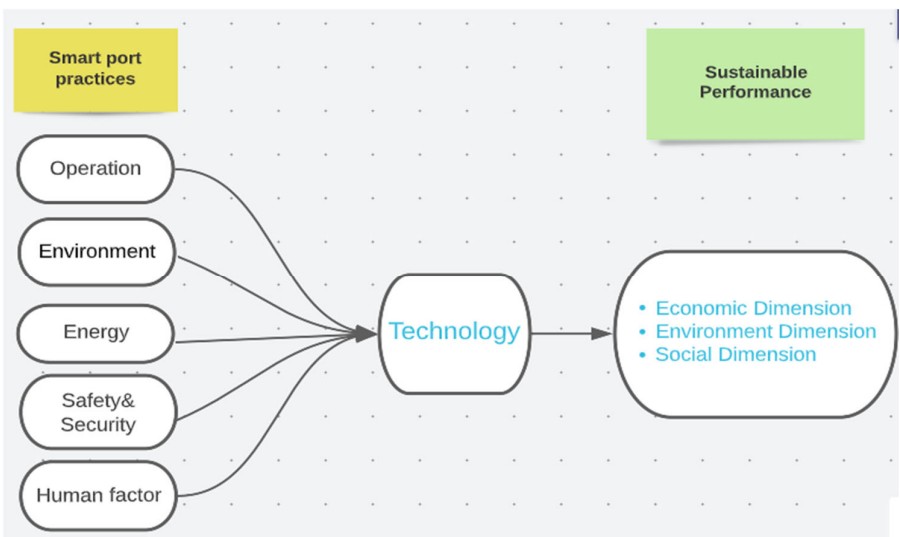

**Figure 2.** The scheme of the research framework. Source: Authors' work.

Consequently, this paper will follow the smart port practices proposed in the above mentioned study, through realizing the different opinions of the core value of smart port practices, technology and port sustainable performance across sectors, as well as identifying the current Egyptian situation by conducting interviews with different stakeholders to identify to what extent there is a level of readiness of adaptability to smart port practices and technology employment in the Egyptian ports in the following factors: operations, energy, environment, safety, and human resources in order to investigate the adaptability of smart port practices, and its influences on sustainable performance. This is followed by a deep analysis through (SWOT) analysis to see the ability of Egyptian ports to adapt to the novel index, and investigate the main obstacles and challenges associated with adaptation.

Then, a focus group will be conducted to conclude some procedures to draw suggestions and recommendations that determine the requirements to adopt the digital transformation towards a sustainable smart port performance.

Figure 3 illustrates the research methodology for investigating the Egyptian ports' level of readiness for applying the smart practices and employing technology to achieve and improve port sustainable performance.

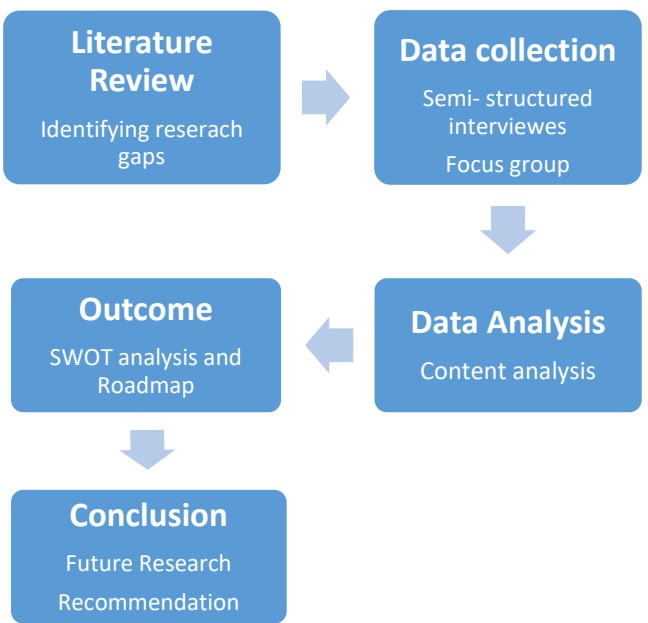

**Figure 3.** Research methodology. Source: Authors' work.

### 3.2. Participants

A group of ten experts (10) were involved in the study (see Table 1). Adopting Thomas and Simerly's [41] approach, the experts were intentionally selected, relying on their positions as decision-makers and their acknowledgment as experts in the field, having experimented with this specific subject and experienced it in their previous work. The population was based on the top-level administrators of maritime-related communities with sufficient years of industry experience.

The interview participants included 10 stakeholders; in terms of government two experts were recruited from the Egyptian Ministry of Transport and two experts from the Egyptian port authority, while in terms of private sector, two experts from shipping operators and freight forwarding companies, and two experts from import and export companies, and finally two experts from the academic field of port management were selected.

Then, the focus group participants included six stakeholders from different sectors: port management, import and export companies, government, and experts in the transport and logistics field. The session was accomplished in two languages, English and the experts' native language, which lessened any language obstacles for the experts.

### 3.3. Data Analysis

The meetings with the interviewees were one-on-one meetings, which lasted for 50 min each, with one interviewer (the author of this study) and one interviewee from different stakeholders. The focus group discussion, which lasted three (3) hours was collected by tape and audio recording. Then, the full recordings were transcribed, and the data were investigated.

This research adopted 'content analysis' for both phases (semi- structured interview and focus group) as the main tool; content analysis is one of the most common procedures used for analyzing qualitative data, by which the researcher investigates in depth what the interviewees referred to mostly and how appropriate their answers were to each

other. In this research, the following steps were taken within the notion of the content analysis method:

(1) Data were collected by audio recording with the permission of the interviewee, and then the recordings were translated to English by the author of this study to guarantee that nothing was forgotten or misunderstood.

(2) After the transcription, the responses were grouped, classified, and compared individually for each inquiry to achieve a deeper insight.

(3) The classified data were emulated to related ideas discussed in previous studies.

(4) The data were analyzed, and the results were revealed in a structured way.

**Table 1.** Participants.

| Number | Organization | Position in the Organization | Years of Experience |
|---|---|---|---|
| 1. | Authority of the Maritime Transport ministry | Associate manager | 15–20 |
| 2. | Marine Department | Manager | 15–20 |
| 3. | DP World Sokhna port | Senior officer | 15–20 |
| 4. | East Port Said port | Manager | >20 |
| 5. | Import and export company | Manager | >20 |
| 6. | Import and export company | Manager | >20 |
| 7. | EIFFA (Egyptian International Freight Forwarding Association) | Secretary General | >20 |
| 8. | Hapag-Lloyd container shipping | Manager | >20 |
| 9. | College of international transport and logistics | Head of department | 15–20 |
| 10. | College of marine engineering | Head of department | 15–20 |

Source: Authors' work.

## 4. Empirical Study

This section attempts to assess the current situation of the Egyptian ports' performance to address the level of readiness for the adaptability of the digital transformation and determine the obstacles and challenges that face them in the adaptation.

### 4.1. Assessment of the Level of Readiness of the Egyptian Ports

Based on all 10 experts who participated in this study, data were collected to know the current state of ports and to specify the extent of adapting smart port practices and technology employment to achieve and improve port sustainable performance in Egypt. The following data were extracted from them to evaluate their level of readiness and adaptability of smart practices and technology in all port activities and identify the main challenges and obstacles for adaptation

4.1.1. Assessing the Current Situation of the Egyptian Port Performance

All the interviewees specified that the government has shown major interest and support to the digital transformation process. This is in line with the vision of Corporate Stack to offer various solutions to help small and medium enterprises (SMEs), and even larger corporations to achieve digital transformation, since President Abdel-Fatah El-Sisi aimed to continue concentrating on monitoring the achievement of the Egyptian Vision 2030, which seeks to reach a diversified, competitive, and balanced economy through the sustainable development framework.

It should be noted that infrastructure in Egypt is realized by the government as a vital driver for the economic development; nearly USD 106.25 million was expended on

infrastructure in the past six years. Public investments were devoted to the transportation and logistics part, which grew by 101% in 2020/2021, compared to the previous years.

Many improvements have been implemented lately; the Alexandria Port Authority provides tractors, with a tensile strength of 40 and 50 tons; additionally, there is the development of the first and sixth area's formation of a security enhancement network consistent with the department of electronic warfare, while Damietta Port Authority builds and delivers an anti-pollution launch boat and pilotage launch boat made of aluminum, as well as the General Authority for Red Sea Ports providing two oil pollution combat units, two solid waste units, and two service launch boats.

The interviewees clarify that a new multipurpose terminal in Alexandria is being developed as a piece of a broader strategy to extend and upgrade the ports of Dekheila and Max, which is anticipated to be completed in 2024 when opening the larger Alexandria Port. Its strategy is to link the Alexandria, Dekheila, and Max ports and integrate them into the larger Alexandria Port, expected to become the largest on the Mediterranean, with 87 terminals expanding over 24.9 km and depths of 20 m. The multipurpose terminal will render imports, exports, and transit goods, as well as increase the percentage of hosting larger vessels and raising the number of containers trading annually from one million to 2.5 million.

### 4.1.2. "Port Sustainable Performance" Concept and Practices in the Egyptian Ports

All the interviewees were familiar with the sustainable port concept; they substantiate the belief that the plan for the transportation sector, which constitutes about 50% of such projects, is to expand public green improvements as a ratio of public investments to 30% in the year 2021/2022, preferring green schemes and gradually retreating from unsustainable projects.

Some interviewees affirmed that going green and using renewable energy is a vital role for Egypt to satisfy its energy needs, increase sustainable economic growth, and create job opportunities through attaining sustainable development goals. Consequently, several recent tenders have drawn strong international attraction and promising proposals, which could further help build up renewable power generation in the coming years. Many renewable energy projects are now under development, imaging the government's determination to turn this vision into reality. Hence, the government's latest aim calls for 20% of Egypt's power generation to be relying on renewables by 2022, and 42% by 2035.

Some respondents highlighted some initiatives and projects that recently took place in Egypt; in the year 2020, the Ministry of International Cooperation declared the beginning of the implementation stage of a Converting Climate Finance Systems with the French Development Agency (AFD). The project seeks to supply long-term loans and technological aid to firms at the cost of USD 182 million, specifically focusing on waste management and transportation.

In addition, Egypt is committed to maintaining life below water and safeguarding marine and shore ecosystems from the pollution that is considered a part of the recently launched presidential initiative through: firstly, the Go Green Initiative, which took place in 2020 via the Ministry of Environment in cooperation with the Red Sea Governorate, the Chamber for Diving and Marine Activities, in addition to the Hurghada Environmental Protection and Conservation Association (HEPCA) for extending the cleaning campaign for the Red Sea through the Red Sea Reserves. Secondly, the Bar Aman Initiative (Safe Shore), which took place in 2021 via the Ministry of Environment and the Governor of Fayoum with collaboration with the Public Authority for Fish Resources Development and the Tahya Misr by providing 42 thousand national fishers with suitable environmentally friendly equipment to support their work, as well as giving them social security and health insurance.

Thirdly, the interviewees ascertained that many other local initiatives have been applied; a further instance of this is when Egypt's Red Sea governorate in 2019 prohibited the use of single-use plastic bags and other objects to lay the foundations for cleaning

up the environment; additionally, Hurghada Environmental Protection and Conservation Association (HEPCA) established a public campaign to increase the awareness about the harmful impacts of plastic pollution on both maritime life and human health.

### 4.1.3. "Smart Ports" Concept and Practices in the Egyptian Ports

All the interviewees were familiar with the concept of smart ports; they agreed that a smart port is an international trend that no one can keep their eyes away from. They also emphasized that the Egyptian road transport has witnessed huge investments and development in terms of being smart in the last few years, and as mentioned earlier that the Egyptian initiatives vision 2035 will be a starting point for the way towards comprehensive development, especially in the maritime sector.

It has been identified that the communication and IT sector in Egypt has perceived an extraordinary precedent where it expanded with a double-digit figure in the period 2005–2010, with an annual average growth of 13.4%, and according to the uncertainty in the political and economic situation, it has been suffering from some drawbacks that have been solved again in reverse, starting from 2016 until now, with a yearly average growth of 13.7%.

They stated that the government of Egypt is one of the first in the world to execute ACI with the usage of block chain technology to reveal unparalleled dedication to technological improvement in the maritime shipping industry and customs clearance technology. CargoX will perform as a gateway for block chain document transfer, involving ACI declaration, bills of lading, and other required original documents. It permits customs authorities to stop depending on declarations from importers. Rather, each document can be effortlessly traced back to its source, straight to the issuer.

It has been confirmed that the new Tahya Masr Multipurpose Terminal will turn Alexandria Port into a regional and global hub for trade and logistics; 88 percent of the construction work has been finalized at this time. They mentioned that the project is being executed with a capacity of 15 million tons annually, with total berth lengths of 2.5 km and depths of up to 17.5 m on an area of 155,000 square meters.

Accordingly, most of them declared that Alexandria Port, Port-Said Port, and Damietta Port are the most appropriate ports in Egypt to apply smart technology in as they are already using many new technologically advanced systems, such as Automated Container Code Recognition (OCR) and Radio terminal data (RTD). By using OCR, the truck carrying the container enters and is scanned by this OCR device, outputting a report telling the situation of the container, the temperature in case of a refrigerator container, and also mentioning if any internal or external damage happens; it also monitors the navigational line of the container and enters the serial number of the container, which includes all data; on the other hand, RTD is a mobile handed device that can locate the container inside the terminal with the cooperation of the EDI system just by entering its serial number, or with additional data like the arrival and departure dates of the container.

There is also a new system that has been operated in 2020 called TOS (Terminal operating system); it clouds the data online, which simplifies container tracking as well as guarantees that there is a space on the terminal for it, whilst also knowing all the container data (type/type of goods/quantity). Moreover, they sustain cargo owners with a tracking system that simplifies the process of tracking their goods online (RFID) that is ready to adapt real-time tracking with the smart technology, and they believed that this will improve the connection and procedures with port administrative, which will raise the efficiency between the port and shipping lines companies and solve any delays, also extending the usage of the solar system. Another issue, from a respondent's point of view, is that Safaga port is one of the most suitable ports after Alexandria port and Damietta Port; the respondent affirmed that among the elements of development is to be a smart port; Safaga is starting to work with Dubai Ports and the Egyptian Ministry of Transport, and it has been agreed that among the phases of development, turning into a smart port will be essential.

### 4.1.4. The Impact of Smart Practices on Port Sustainable Performance

All of the interviewees confirmed that smart practices influences port sustainable performance; they agreed that innovating in sustainability and technology will positively affect the entire port's overall performance by using the resources in an efficient and effective way without any wastes and finding alternatives to the harmful resources that affect the environment, substitute to the high cost resources, and increase efficiency, safety, and security through intelligent practices across all the logistic processes within the port.

One of the reasons why respondents think this is the case is based on the COVID-19 pandemic in 2019, which influenced ports geographically and sectorially; the pandemic assisted to remind ports that they are more interconnected than segregated. Consequently, international cooperation and integration is essential, not a luxury.

The pandemic practically trained the world in that sustainable development should be the greatest objective and that normal trading depending on human resources jeopardizes development and puts people and the planet at risk, in addition to the building of deeper terminals and logistics areas and accelerating the pace of digitalization, which will enhance the loading and unloading process, reduce the waiting times of ships, improve the skills of customs employees, and enable customers to finish their papers in one place, which will reduce the cost of custom clearance.

Respondents also pointed out that not only do smart port business models influences port sustainable performance and its dimensions (social, economic, and environment) but also adopt the growth markets included within societal sustainable requirements. Digitalized cargo data prevents unneeded physical moves. Better synchronization decreases emissions, travel time, cost, and congestion. Ports that are now often recognized to be on the wrong flank of subjects related to climate, globalization, or mobility, can change their image through sustainable innovation.

### 4.1.5. The Main Expected Benefits from Adapting the Sustainable Smart Performance in the Egyptian Ports

Respondents interviewed in this study seem generally significantly positive towards the concept of digitalization and adapting smart practices. They emphasized that sustainable transmission, receipt, and the response of information needed for the arrival, stay, and departure of vessels, individuals, and shipment, including information and declarations for customs, immigration, port, and security controls, can be achieved through electronic data exchange and smart practices

Most of the respondents in this study also believe that adapting sustainable smart ports can ease the sharing of port and berth master data for the just-in-time process in ships and ideal resource positioning of vessel services and suppliers, logistics providers, cargo handling, and clearance, accordingly saving energy and enhancing safety, in addition to reducing costs and emissions.

Furthermore, its applications have the prospect to solve continuous difficulties in Egypt: red tape, bribes, the incidence of corruption, low transparency and accountability, and high transaction costs could be decreased by the distribution of smart practices in port society.

Another issue from respondents' point of view for the adaptation is that it will increase the standard of the port to compete internationally, and according to the geographical location of Egypt, this will expand the services and the scope of the port coverage not only to serve neighboring countries but also to be the most important global logistics hub.

In addition, it leads to adopting a low-impact usage of resources, fuel, and electricity consumption, becoming more promising neighbors with neighborhoods close to ports and terminals, tackling air quality, dust, sediment and water quality, and ballast water management, in addition to other possible irritation problems such as traffic, noise, and safety matters, and demonstrating consistency in measurement and documenting from all points in the logistics and transport.

### 4.1.6. The Main Barriers and Obstacles to Implement the Sustainable Smart Port Performance in the Egyptian Ports

As highlighted above, all the interviewees confirmed that it would be very helpful if the Egyptian ports adapt sustainable smart practices, and that this would lead to smoothness in the trading movement in Egypt. However, based on the interview analysis, the interviewees summarize the barriers of adapting smart port practices in Egyptian ports in three main points:

(a)   Lack of technological solutions
(b)   Socio-economic, organizational and regulatory
(c)   Economic and political issues

Adapting smart ports in Egypt is challenging because of the weak infrastructure and the huge amount of funds required to maintain and enhance. Cost and resistance to change are the biggest barriers that face Egyptian ports to transform into sustainable smart ports, alongside the lack of knowledge and experience, especially in the operation field. However, the Egyptian port authority focuses on performing training plans and programs to prepare and develop human resources, in coordination with training institutes and centers internally (Port Training Institute) and externally (scholarships and trainings Sweden, Japan, China, India, South Korea, and Thailand).

One of the respondents mentioned that implementation will cost a lot in the short run, while in the long run, it will cover its costs by generating a huge amount of revenues. Respondents also pointed out that custom clearance in Egypt was considered for a very short time as a great challenge because it was 100% un-automated and it is the first operation that customers must do when they receive their shipment, at the beginning of applying the ACI; the custom clearance workers did not accept to change their process because they thought that the application is not unified and not standardized and because of their resistance to change, but after a while, all things settled and changed. From the respondents' point of view, if most of those challenges have been solved, then ports can manufacture sustainable smart ports.

### 4.2. Obstacles and Challenges for Adaptation

Based on six (6) of the experts who participated in this study, data were collected to identify the main challenges and obstacles of adapting smart port practices and technology in the Egyptian ports and the following data were extracted from them to identify the procedures and suggested actions that might facilitate the process of adaptation in the Egyptian ports.

### 4.2.1. The Main Obstacles and Barriers to Adapt the Smart Port Practices and Technology Employment

The interviewees revealed that the fundamental challenges nowadays confronting investment in this sector are the lack of skilled resources to drive all these digital transformation projects and the lack of government funding, particularly as the pandemic continues to hit the nation, while others stated that humans will take time to learn but that training and seminars can help overcome this problem.

In addition, Egypt is facing a number of financial barriers, including government debts and budget deficits; interviewees observed that the Egyptian container ports are trying to improve their current position in the logistics index, and they are carrying out more efforts in developing the infrastructure of their ports in accordance to the government plans; however, some projects face a lot of problems, such as lack of funding and integration.

Moreover, others stated that Monitoring and Maintenance technology is considered a big problem as there is no unified system that shares all the required data and connects container terminals, shipping lines, customs clearances, and port authorities with each other. In addition, governmental permissions and decisions take a long time in Egypt and thus must change their approval routine; they must be quicker in making decisions and approvals with the smart and modern system.

### 4.2.2. The Main Requirements Needed to Adapt Smart Port Practices and Technology Employment

There are many requirements that must be applied to adapt smart port practices: first, the infrastructure in Egypt needs innovation and support from the shipping lines and the Egyptian government to be able to apply smart port practices.

Second, there must be a technological system to connect container terminals, shipping lines, customs clearances, and regular authority with each other to apply smart port practices in Egypt and to have a cloud in order to consequently be connected by the internet and networks; additionally, ports must adapt programs and smart clouds to increase integration within the port.

Third, new developments should be applied in ports and designs to utilize the latest technological advances to transform into SSP; they believe that it is vital to create data, voice, and security networks that are connected to control, safety, and security systems within ports, connecting them with each other in order to allow the operational, maintenance, and security teams to have full port visibility.

Other experts stated that in order to adapt sustainable smart ports in Egypt, it is required to have energy management certificates or arrangements according to any standards (ISO 50001, etc.), number of safety and security arrangements and certificates, certificates in maritime environmental management, operator of lifting equipment training, cargo coordinator training, planner training and port worker training program in safety and health. It is also important to provide services with greater reliability and higher standards of quality, security, safety, financial sustainability, resource protection, environmental protection, and community participation. Besides stimulating investments, Egypt should improve its ranking in the three international indicators of global competitiveness, doing business, and the macroeconomic environment.

In addition, it is important to build a strong Wi-Fi internet connection that will provide remote site status, equipment monitoring, alarms, and notifications; one of the respondents stated that regular maintenance should be carried out to avoid failure, error, and replacement costs.

Other experts suggest constructing ships with modern innovations, using the renewable energies system for the ships' designs, moving towards integration intermodal transport techniques, decreasing emissions, spreading over the tracking and tracking technologies, focusing on more automation and integration of data, promoting investment in technology, enhancing the environmental and waste management systems, adopting the corridor management strategies, and developing and implementing sustainable energy action plans, which will facilitate the vision of adopting SSP practices.

### 4.2.3. Suggestions and Comments Added by Stakeholders

Interviewees suggested encouraging the implementation of the following technologies will improve health security in port environments, allowing port and marine employees, contractors, and the vessel crew to cooperate and interact in the securest possible condition, and that this can be achieved by raising awareness, avoiding incomprehension, fostering best practices, and standardization regarding how port communities can apply emerging technologies; for instance, artificial intelligence, advanced analytics, internet of things, digital twins, robotics process automation, autonomous systems, block chain, virtual reality, and augmented reality.

It has become clear that after the COVID-19 pandemic, the world has realized that there is a critical need for inter-governmental organizations, governments, and industry stakeholders dealing with maritime trade and logistics to come together and accelerate the speed of digitalization so that port communities can use electronic commerce and data exchange, in observation with all appropriate contractual and regulatory obligations.

The increasing volume of data, the need for mobility in logistics, and the exchange of information also increase the demand for data security and data protection in maritime logistics to stop the manipulation of sensitive systems; consequently, all players in the

maritime supply chain will have to provide the best attainable protection to defend their data against unauthorized access and any kind of abuse by cloud-based user approaches, access management, and data backup.

By going smart, connectivity and automation will help decrease environmental footprints of the port industry together with smart transport systems, which will reduce $CO_2$ emissions. New requirements on efficiency, security, and the environment demonstrate where the coordination of industry and policy is necessary to incorporate smart port practices among all the logistic operations within the port.

Adopting smart port practices and technology employment in the Egyptian ports linked to port sustainable performance is essential. However, it is not possible to suddenly transform into a smart port; this transformation must take place in stages and over several years.

## 5. Discussion and Findings

This paper has four key aims: firstly, using primary and secondary data to evaluate the Egyptian current performance and situation; secondly, analyzing and assessing the level of readiness and adaptability to smart practices in the Egyptian ports context; thirdly, identify the main obstacles and challenges; and finally, presenting recommendations for policymakers.

After carrying out this study, the analysis of the primary data collected through several interviews and the focus groups and based on the above-mentioned various relevant authorities from the public and private sectors and experts in the field concurrently confirmed that smart port practices and technology employment have a positive influence on port sustainable performance. Several findings were reached. First, to what extent there is a level of readiness of adaptability of smart port practices and technology employment to achieve and improve port sustainable performance. Second, the main obstacles and challenges associated with the adaptation, which can be summarized by SWOT analysis. Third, based on the focus group, developing some procedures that determine recommendations and suggestions regarding the obstacles associated with adaptation.

### 5.1. SWOT Analysis

SWOT analysis is a tool used for verifying and evaluating an organization and its environment. SWOT is used to assess the internal strengths and weaknesses, external opportunities, and threats [42].

Based on the previous work, research problem, and gaps, the research specified that the way of adapting sustainable smart port practices and technology employment in the Egyptian ports is to evaluate their current situation and level of readiness by SWOT analysis in order to know their possibilities, opportunities, and to determine the weaknesses points and external threats. Data were collected and summarized from the above interviews investigation, as illustrated in the figure below (Figure 4).

### 5.2. Strengths

Regarding the Egyptian Vision 2035, one of the major strengths is the governmental support that offers various solutions to help (SMEs) and large corporations achieve digital transformation, which seeks to reach a diversified, competitive, and balanced economy through the sustainable development framework. In addition, many improvements have been implemented lately in the Alexandria Port Authority, such as developing the first and sixth area's formation of a security enhancement network and providing tractors, while the Damietta Port Authority delivered an anti-pollution launch boat and pilotage launch boat made of aluminum.

In addition, some initiative projects have recently taken place in Egypt: in 2020, the Go Green Initiative extending the cleaning campaign for the Red Sea through the Red Sea Reserves. Further, in 2021, the Bar Aman Initiative provided 42 thousand national fishermen with suitable environmentally friendly equipment to support their work, as well

as giving them social security and health insurance and life below water and safeguarding marine and shore ecosystems from the pollution project.

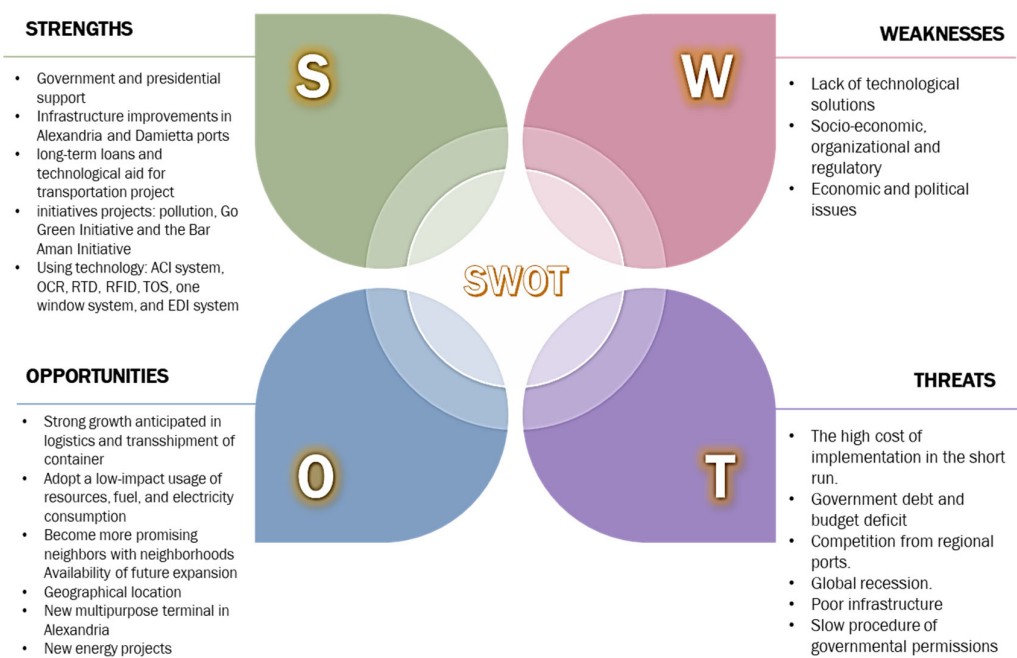

**Figure 4.** SWOT analysis of the adaptation of sustainable smart port practices in Egyptian ports. Source: by author.

More importantly, the communication and IT sector in Egypt has perceived an extraordinary growth from 2016 until now, with a yearly average growth of 13.7% [43]. Many technological improvements in the maritime shipping industry have been executed lately, such as the one window system and EDI system, ACI with the usage of CargoX block chain technology for customs, Automated Container Code Recognition (OCR) TOS (Terminal operating system), and Radio terminal data (RTD) for containers, as well as (RFID) for the process of tracking goods online.

### 5.3. Weaknesses

Adapting smart ports in Egypt is challenging and has several weaknesses: first, the weak infrastructure, unsatisfactory Wi-Fi internet connection, and poor IT logistics, which requires a huge amount of funds to enhance. Moreover, social acceptability of the new system and the resistance to change by the employees and the eco-system, as well as a lack of knowledge, experience, and skilled resources, especially as the level of higher education and training is low. Therefore, Egyptian ports somewhat suffer from a shortage of skilled and experienced logistics professionals. Further, there is a lack of integration, regular monitoring, and adjustable maintenance technologies as there is no unified system that shares all the required data and links the ports' different departments to be capable of managing their traffic and trade flows and managing the transportation of goods.

### 5.4. Opportunities

The Egyptian vision 2035 initiative is considered one of the main opportunities on the way towards comprehensive development in the maritime sector, as the plan for the transportation sector in the upcoming year is to expand public green improvements as a ratio of public investments by 30%, with a strong growth anticipated in the logistics and transshipment of container trade in the Mediterranean district.

The initiative aims to improve the competitive situation of Egyptian ports among others, decrease energy consumption and associated greenhouse gas (GHG) emissions, and adopt a low-impact usage of resources, fuel, and electricity consumption.

According to the Egyptian geographical location, its borders are with North African countries and the Middle East counties; improving the country's inland infrastructure and multimodal connections with those countries, the port will improve its cargo throughput. The adaptation also will increase the standard of the ports to compete internationally, which will expand the services and the scope of the ports' coverage to not only serve neighboring countries but also to be the most important global logistics hub.

One of the most important opportunities is the availability of future expansion. Egyptian ports have a great expanse of land for the ability to expand in order to cope with future capacity need.

The new multipurpose terminal in Alexandria, which will be completed in 2024 and is considered a piece of a broader strategy to extend and upgrade the ports of Dekheila and Max, is expected to become the largest in the Mediterranean, with 87 terminals expanding over 24.9 km and depths of 20 m. This new terminal will raise the number of containers trading annually from one million to 2.5 million. The project is being executed with a capacity of 15 million tons annually, with total berth lengths of 2.5 km and depths of up to 17.5 m on an area of 155,000 square meters. In addition, through the collaboration of Safaga with Dubai, it has been agreed that among the phases of development, turning into a smart port will be essential.

Several recent tenders have drawn strong international attraction and promising proposals, which could further help build up renewable power generation in the coming years. Many renewable energy projects are now under development, imaging the government's determination to turn this vision into reality. Hence, the government's latest aim calls for 20% of Egypt's power generation to be relying on renewables by 2022, and 42% by 2035.

### 5.5. Threats

Egyptian ports face a number of threats. First, the high cost of implementation in the short run and the high rate of alternative fuels, which is considered a financial barrier, including government debts and budget deficits; this also is impacted by the lack of government funding, particularly as the pandemic continues to hit the nation. In addition, the Egyptian governmental permissions and routine process, which takes a long time to make new decisions, as well as lack of efficient production approaches to raise competitiveness among others, while Egyptian ports simultaneously encounter severe competition from the main hub ports in the Mediterranean area.

### 5.6. Key Challenges for Egypt to Adopt Digital Transformation

The SWOT analysis that assesses the current situation investigated that the Egyptian ports face many obstacles. A focus group was formed with appropriate parties to investigate those obstacles and the most important procedures for adopting digital transformation, draw findings and suggestions in light of the above, and propose some procedures to enhance the adaptation of smart practices and technology employment in Egyptian ports, which will improve their sustainable performance.

By asking the participants about the most important obstacles that Egyptian ports may face while adopting smart practices and technology employment, the participants explained that although there are huge efforts exerted from the beginning of the automation and digital transformation efforts, Egyptian ports are still far from reaching the expected objective of adapting to smart ports due to many challenges.

To encounter these numerous obstacles, the participants in the study categorized them in this regard. First, the internal obstacles, which mean the challenges that are related to the control of ports processes. Second, external obstacles, which means challenges that are beyond the control of the ports but still negatively affect the process of adaptation. It is to be noted that there are no obvious lines between the two categories; some of the obstacles are of mixed nature.

A.　Internal obstacles:

- There is a lack of experts in policy formulation, strategy development, and/or project management within the port practices. There is no awareness of the significance of building the capacity of employees or experts and providing them with the required skills to equip expertise with the technical skills needed for smart adaptation.
- Lack of education and capacity improvement is considered one of the obstacles to achieving sustainable smart port performance and improving strategic plans.
- The lack of non-availability of information; current work systems do not allow the supply of authentic information to create databases that simplify the process of adaptation. The absence of collaboration is one of the most important obstacles; it desolates the opportunity of collecting the dispersed efforts under the umbrella of one unified vision, which properly utilizes the available resources to attain the best outcomes.
- Lack of permanent monitoring and adaptive maintenance technologies, which affects the efficiency of production and processes.

B.　External obstacles:

- Water scarcity is regarded a significant obstacle in Egypt to attaining sustainable development goals; this is due to global warming, disposal of wastes in the water, and the establishment of the Grand Ethiopia Renaissance Dam (GERD).
- The difficulties concerning the availability and sufficiency of financial resources are extremely rooted in the history of port development.
- Bureaucratic and regulative inefficiency in public sector.
- Increasing the global $CO_2$ emissions originating from sea transport.
- Ship energy efficiency is another obstacle because of the future Egyptian context of the rarefaction of oil and power if compared with the other States in the Middle East.
- The high unemployment rate, specifically among educated people, because of the absence of linkage between the education system and market needs.

Table 2 represents a review of the results from the previous analyzes (and in light of the successful experiences that were reviewed), the suggestions submitted by the participants in the study, the scheme of the procedures that summarizes the challenges derived from interviews, literature reviews, suggestions and recommendations by supply chain experts and industrial specialists regarding the adoption of digital transformation in moving towards a sustainable smart port performance.

**Table 2.** Suggested some procedures to adopt the digital transformation towards a sustainable smart port performance.

| Obstacles | Suggestions |
|---|---|
| Economic Dimension | |
| Lack of integration | Unifying system that shares all the required data and connects container terminals, shipping lines, customs clearances, and ports authority with each other. Moving towards integration intermodal transport techniques. Preparing an integrated electronic database. |
| Securing adequate power production | Boosting innovation in the energy sector. Adopting an inclusive program to foster innovation and knowledge culture. |
| Poor IT logistics | Promoting investment in technology. Building strong Wi-Fi internet connection. Using remote site status and monitor equipment. |
| Outdated infrastructure | Developing communication and technological infrastructure innovation. supported from the shipping lines and the Egyptian government. Constructing ships with modern innovation. |

**Table 2.** *Cont.*

| Obstacles | Suggestions |
| --- | --- |
| Environmental Dimension ||
| Waste disposal | Improving the efficiency of the solid-waste management system and supporting its sustainability. Creating a system for disposal of hazardous wastes. Increasing the awareness on the efficiency of saving coastal and marine areas. |
| Limited environmental awareness | Focusing on more automation and integration of data. Enhancing the environmental and waste management systems. Adopting the corridor management strategies. Developing and implementing sustainable energy action plans. |
| Lack of monitoring and maintenance technology | Adapting regular maintenance programs and smart cloud-based user approaches, access management, and data backup. Creating data, voice, and security networks, and utilizing alarms and notifications. Spreading over the tracking and tracking technologies. |
| Social Dimension ||
| Lack of skilled resources, technical expertise, and knowledge on new technologies | Performing training plans and programs. Improving the human resource management system. Training and capacity building for planning and monitoring units in different departments. |
| Low level of higher education and training | Improving the quality of education and training skills. Providing professional certificates with collaboration with accredited entities to workers on the technical and administrative level to cope with the technological growth in various fields. |
| Social acceptability of the new system | Raising awareness. Avoiding incomprehension. Fostering best practices, and standardization on how ports' communities can apply emerging technologies. |

Source: Authors' work.

## 6. Conclusions

This study investigated to what extent the Egyptian ports could apply the smart practices and employ technology to achieve and improve sustainability performance through identifying the current situation of the Egyptian ports' performance and investigating the level of readiness and adaptability to smart port practices and technology employment in the Egyptian ports across five factors–operations, energy, environment, safety, and human resources–and their influence on achieving and improving sustainability performance. Furthermore, a strength, weakness, threat, and opportunity (SWOT) analysis was performed to see the ability of the Egyptian ports in adapting smart port practices and technology employment, investigate the main obstacles and challenges associated with adaptation, and to conclude with some procedures that determine the requirements to adopt the digital transformation towards a sustainable smart port performance.

The empirical study of Egyptian seaports showcased that the digitalization and smart port practices can be successfully adapted. However, as a result of the conducted research, the research investigated that the difficulties associated with Egyptian smart port adaptation are: there are many old ports that came into service a few decades ago with a lack of monitoring and maintenance technology; these ports have a scarcity of financial support and are under the management of outdated methods. Another problem is the weak consciousness of environmental protection and energy-saving, which results in limited resources assigned to perform systematic and comprehensive investigations when conducting smart port planning and design, in addition to slow governmental permissions and routine, lack of awareness, lack of skilled resources, the lack of government funding, and low educational level. In addition, the incompleteness of vital evaluation benchmarks for the development of smart ports causes certain blindness in the smart port adaptation and negatively affects the sustainable smart port performance in their social, environment, and economy aspects.

The usage of technology has become essential to facilitate process improvements across port logistics. Many ports around the world recognize that their future is based not only on infrastructure expansion but also on adopting smarter strategies and smooth integration of the port community.

Adaptation of sustainable smart ports in Egypt will be very helpful and will solve congestion in ports and increase efficiency of time, decrease the consumption of fossil energies, lower the environmental effects of the vessels, and fix the safety/security challenges faced by ships, which will lead to fluidity in the trading motion in Egypt.

To fulfil the Egyptian vision of transforming ports, powerful industry leadership and a change in mindset are mandatory as all data need to be linked between ports' different departments to enable them to manage their traffic and trade flows and the transportation of goods.

By going smart, connectivity and automation will help decrease environmental footprints of the port industry, as will smart transport systems that reduce $CO_2$ emissions. Smart ports maximize the use of space, time, money, and natural resources efficiently and effectively, assisting in greater operational and energy efficiency, promoting safety and security, and enhancing environmental sustainability, which will improve the overall performance of the port.

This paper contributes to the existing literature of smart port practices and technology employment in the Egyptian ports. It also serves as an exploratory study to apply the idea of smart seaport adaptation. This work will promote the development of sustainable performance and smart theory in the port industry as the use of smart technologies ensure the sustainable performance of maritime transportation and the transport system as a whole. Furthermore, the evaluation results generated from the empirical study of port cases in Egypt demonstrate their advantages and deficiencies in terms of the smart seaport adaptation. Thus, the procedures and flexible methods presented in this paper will be applicable for evaluating the development level of smart ports and be capable of providing useful insights and a guide for port authorities and stakeholders to improve management.

## 7. Future Work

Since there is limited previous work in the literature that attempts to enhance the smart port practices and technology in Egypt, and after concluding the analysis and interpretation of the findings from interviews and focus group, this research encounters some further limitations, followed by some suggestions for further review papers are as follows:

- The research focuses on applying empirical study through one country, particularly Egypt. However, the empirical research can be extended. Further research can test the implementation of smart port practices and technology employment within the Egyptian context through developing roadmap that can be generalized to evaluate ports in any country based on the relative importance weights assigned to the key selected indicators.
- The research is applied only on one country; however, a comparative study between different developing countries will determine if they face the same obstacles and barriers.
- This study was a qualitative study; however, further research can conduct a survey to test the adaptability regarding the applicability of the smart practices and technology employment, and then test its influences to improve port sustainable performance in the Egyptian seaport context.

**Author Contributions:** Conceptualization, A.O. and S.E.G.; methodology, A.O. and M.K.; writing—original draft preparation, A.O.; writing—review and editing, A.O. and S.E.G.; supervisions' and M.K. All authors have read and agreed to the published version of the manuscript.

**Funding:** This research received no external funding.

**Institutional Review Board Statement:** Not applicable.

**Informed Consent Statement:** Not applicable.

**Data Availability Statement:** The data presented in this study are available on request from the corresponding author. The data are not publicly available due to privacy reasons.

**Conflicts of Interest:** The authors declare no conflict of interest.

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
