# Peer review of "Investigating the Influences of Smart Port Practices and Technology Employment on Port Sustainable Performance: The Egypt Case"

_sustainability, doi:10.3390/su142114014_

Round 1

Reviewer 1 Report (Previous Reviewer 2)

The use of smart technologies will allow, in the context of changes in global logistics processes, to ensure the sustainability and safety of maritime transport and the transport system as a whole, which is why the topic proposed by the authors of the article is undoubtedly interesting and relevant. The content of the article as a whole adequately reflects the formulated problem, and its title and keywords correlate with the text. The abstract describes the essence of the article, briefly outlines the issue state, research methods, and results. The authors also cite limitations and potential beneficiaries of the proposed activities.

In the introduction, the authors give a brief overview of the issue state, indicate the purpose and objectives of the study, and also give the structure of the article. Section 2 provides an overview of the literature on smart ports and sustainability, characterizing Egyptian ports, as well as studies on the level of digitalization and digital initiatives of commercial ports. Section 3 is devoted to describing the methods that the authors used to conduct the study. Section 4 presents an analysis of the research results based on its objectives. Section 5 is devoted to presenting the conceptual framework and discussing the results obtained. In section 6 - "Conclusions" the authors summarize the results obtained and draw conclusions on the work. Section 7 outlines the limitations of this study, as well as possible directions for future research.

The article is relevant to the topic it explores and presents, the article is prepared in accordance with the instructions for the authors. In our opinion, the article corresponds to the theme of "increasing the intelligence of maritime transport and logistics systems" and corresponds in type to the Preliminary study.

 Notes:

1. There are problems with the numbering of the figures (Figure 1 is immediately followed by Figure 4, and "Figure 5" is called 2 different images).

2. There are also problems with the structure of the article (after section 4.2 comes the number 4.7, then again 4.7 and then 4.9, and in section 5 there is no subsection 5.1, but immediately comes 5.2).

3. There is a note on the design of tables - the name of table 2 is located under the table itself.

Author Response

Response to Reviewers

Cover Letter

Journal of Sustainability

Dear Editor,

Subject: Submission of revised paper “Investigating smart port management and technology employment influences on port sustainability performance: An empirical study on Egypt” Manuscript ID: sustainability- 1961424

Thank you for giving us the opportunity to submit a revised draft of the manuscript “Investigating smart port management and technology employment influences on port sustainability performance: An empirical study on Egypt” for publication in the Journal of sustainability. We appreciate you and the reviewers for your precious time in reviewing our paper and are grateful for the insightful comments on and valuable improvements to our paper. Your valuable and insightful comments led to possible improvements in the current version. The authors have carefully considered the comments and suggestions made by the reviewers and tried our best to address every one of them.

We hope the manuscript after careful revisions meet your high standards. The authors welcome further constructive comments if any. Below we provide the point-by-point responses to the reviewers’ comments and concerns. All modifications in the manuscript have been highlighted in red. All page numbers refer to the revised manuscript file with tracked changes.

 Sincerely,

Name: Alaa Othman, PhD. alaa.abomousa@gmail.com

The University of Maribor, Faculty of Logistics

Reviewer 1

[General Comment]: The use of smart technologies will allow, in the context of changes in global logistics processes, to ensure the sustainability and safety of maritime transport and the transport system as a whole, which is why the topic proposed by the authors of the article is undoubtedly interesting and relevant. The content of the article as a whole adequately reflects the formulated problem, and its title and keywords correlate with the text. The abstract describes the essence of the article, briefly outlines the issue state, research methods, and results. The authors also cite limitations and potential beneficiaries of the proposed activities.

In the introduction, the authors give a brief overview of the issue state, indicate the purpose and objectives of the study, and also give the structure of the article. Section 2 provides an overview of the literature on smart ports and sustainability, characterizing Egyptian ports, as well as studies on the level of digitalization and digital initiatives of commercial ports. Section 3 is devoted to describing the methods that the authors used to conduct the study. Section 4 presents an analysis of the research results based on its objectives. Section 5 is devoted to presenting the conceptual framework and discussing the results obtained. In section 6 - "Conclusions" the authors summarize the results obtained and draw conclusions on the work. Section 7 outlines the limitations of this study, as well as possible directions for future research.

The article is relevant to the topic it explores and presents, the article is prepared in accordance with the instructions for the authors. In our opinion, the article corresponds to the theme of "increasing the intelligence of maritime transport and logistics systems" and corresponds in type to the Preliminary study.

Thank you for the comments and suggestions on our manuscript entitled. We appreciate the suggested modifications and have revised the manuscript accordingly. We hope the manuscript has been improved accordingly.

 Notes:

  1. There are problems with the numbering of the figures (Figure 1 is immediately followed by Figure 4, and "Figure 5" is called 2 different images).

We thank the reviewer for pointing out this typo, which has now been corrected

  1. There are also problems with the structure of the article (after section 4.2 comes the number 4.7, then again 4.7 and then 4.9, and in section 5 there is no subsection 5.1, but immediately comes 5.2).

The author revised all numbering and structure of the manuscript, and all modifications have been made

  1. There is a note on the design of tables - the name of table 2 is located under the table itself.

The name of table 2 has been moved before the table

Reviewer 2 Report (Previous Reviewer 3)

Dear Authors,

Investigating smart port management and technology employment influences on port sustainability performance: An empirical study on Egypt

Suggestion to reduce and improve:

Smart port management and technology influence on port sustainability performance: The Egypt case.

Include the number of interviews in the abstract

“this present study aims to investigate to what extent the developing countries could apply smart practices and employ technology to achieve and improve sustainable performance through conducting an empirical study on Egypt”

What technology? From Smart port?

Is the purpose to measure and stusy for developing countries? Why can the case of Egypt be generalized? The objective is to study the case of Egypt, not knowing how other developing countries can apply smart port practices.

Are smart practices or smart port management and technology?

Is it sustainable performance or port sustainability performance?

Must keep the concepts the same in all text.

“adaptation of digital transformation towards a sustainable smart port development”

Do you intend to achieve sustainable smart port development or port sustainability performance? What is the endogenous variable anyway?

Please clarify as are different concepts.

If “Based on the above discussion, it has been proved from previous studies that the technology variable employs and effects the relationship between smart port practices and sustainable performance, which enhance and improve the overall port performance. Different approaches, models and measures have been proposed by many researchers to employ technology to enhance sustainable performance which will be illustrated in next subsection”, what is the GAP refered by authors that this study is filling?

“that there is still a need for an integrated performance measurement system capturing the relationship between smart  port, and sustainability while employing technology”

Is technology an exogenous explanatory variable or a moderating variable in the relationship under study? Clarify the role of this variable. Do we have smart port management technologies and practices as causal variables in the relationship or is one moderator or mediator?

“this paper will follow the smart port practices proposed in the above mentioned study [7] the drivers to sustainable port performance”

Now technology disappears and management becomes practices. It has to be very well defined and keep the concepts. They are different concepts. Are they management or operational practices, cultural or organizational, technological? To clarify?

Sustainable port performance or port sustainability performance? Are completely different concepts. It is one thing to have an overall sustainable long-term performance in the port and quite another to have a good performance in terms of economic, social and environmental sustainability specifically.

"this research is trying to fill this gap while testing, at the same time, the applicability factors instead of theoretical facts. " The facts are theoretical?, the research of theoretical hypotheses can be based on case studies or on large samples, quantitative, qualitative data and varibles, etc. please correct.

RQ3 - please format the point

Materials and Methods – I don't see any question that leads to confirm the hypothesis that smart ports management and technology can affect port sustainability performance. Were there more questions?

"identify the main challenges and obstacles for adaptation" – It is not the aim of the study.

Authors must define well which study variables are explanatory, moderating, explained and the causal model to be proven. Make a table with the variables and which authors used them previously.

What are smart port management and technology? What are its subvariables based on the literature?

What is the concept of port sustainability performance, what are its variables?

How do the first ones influence the economic, social and environmental aspects?

“Egyptian current performance and current projects” – 

sustainable performance or port sustainability performance?

Why is SWOT analysis relevant for the study goals? Should this part be in the study? Why? No add value to prove the hypothesis. The hypotheses have to be revealed clearly and the conclusion for each one must be shoed, includind for each variable.

Author Response

Response to Reviewers

Cover Letter

Journal of Sustainability

Dear Editor,

Subject: Submission of revised paper “Investigating smart port management and technology employment influences on port sustainability performance: An empirical study on Egypt” Manuscript ID: sustainability- 1961424

Thank you for giving us the opportunity to submit a revised draft of the manuscript “Investigating smart port management and technology employment influences on port sustainability performance: An empirical study on Egypt” for publication in the Journal of sustainability. We appreciate you and the reviewers for your precious time in reviewing our paper and are grateful for the insightful comments on and valuable improvements to our paper. Your valuable and insightful comments led to possible improvements in the current version. The authors have carefully considered the comments and suggestions made by the reviewers and tried our best to address every one of them.

We hope the manuscript after careful revisions meet your high standards. The authors welcome further constructive comments if any. Below we provide the point-by-point responses to the reviewers’ comments and concerns. All modifications in the manuscript have been highlighted in red. All page numbers refer to the revised manuscript file with tracked changes.

 Sincerely,

Name: Alaa Othman, PhD. alaa.abomousa@gmail.com

The University of Maribor, Faculty of Logistics

Reviewer 2

[General Comment]: Investigating smart port management and technology employment influences on port sustainability performance: An empirical study on Egypt

Suggestion to reduce and improve:

Response: We would like to thank the reviewer for his thoughtful and valuable comments and efforts toward improving our manuscript. In the following, we highlight the general concerns of reviewers that were common and our effort to address these concerns carefully, we hope the newly updated manuscript has been improved accordingly

Here is a point-by-point response to the reviewers' comments and concerns.

Reviewer 2 Comments

Author Response

Location of Response in Revised Manuscript

1.      Smart port management and technology influence on port sustainability performance: The Egypt case

We agree with the reviewer that this would be better. kindly find the updated title (Investigating the influences of smart port practices and technology employment on port sustainable performance: An empirical study on Egypt: The Egypt case)

Section: Title

Page: 1

2.      Include the number of interviews in the abstract

As suggested by the reviewer, kindly find the updated abstract to be more informative by adding the number of interviewees

Section: Abstract

Page: 1

  1. “this present study aims to investigate to what extent the developing countries could apply smart practices and employ technology to achieve and improve sustainable performance through conducting an empirical study on Egypt”

What technology? From Smart port?

We agree with the reviewer that this seems to be misleading and have amended and modified this section

Change: adding paragraphs illustrating which type of technology

Section: 2.2

Page: 7

  1. Is the purpose to measure and study for developing countries? Why can the case of Egypt be generalized? The objective is to study the case of Egypt, not knowing how other developing countries can apply smart port practices.

We thank the reviewer for pointing out this and 2 paragraphs have been added on page 10 to illustrate that and Egyptian case has been examined as an illustration of the principals involved in smart port competition and many modifications throughout the manuscript have been made

The smart port is essential and challenging all over the world, especially for developing countries; many developing countries in East Asia and the Middle East have focused on the smart port/ city approach as a contemporary urban development policy to meet the challenges and issues of urban management in developing countries. Egypt is one of the developing countries that has gone through transitional stages and faces many challenges concerning this transformation, thus this research aims at investigating to what extent the Egyptian ports could apply smart practices and employ technology to achieve and improve sustainable performance

Accordingly, the whole manuscript has been revised and those parts have been modified and it has been clarified in the following sections:

•                     Section: Abstract, Page: 1

•                     Section: 1, Page: 3 & 4

•                     Section: 2.3, Page: 8&10

•                     Section 3: page: 11

•                     Section: 6, Page: 27

•                     Section: 7, Page: 28

  1. Are smart practices or smart port management and technology?

Smart port practices, as the research investigates the level of readiness and adaptability through the five dimensions that were already measured and analyzed before in previous studies mentioned. One of the most comprehensive studies dedicated to the majority of the previous literature, tackling and measuring the variables used in detail has been mentioned and adopted as a guideline for our study and mentioned specifically on page number 8, by explaining the index dimensions, findings, limitations, and further research of this study. Our study adapts those dimensions in the Egyptian context and test to what extent this can be applied in the Egyptian ports, investigated the challenges and obstacles of adaptation then suggest procedures to overcome them

Change: unify all the concepts to be smart port practices

The whole manuscript

  1. Is it sustainable performance or port sustainability performance?

Must keep the concepts the same in all text.

The term has been unified in the whole manuscript to be (port sustainable performance) as it’s the main concept, it deals with the measurement and disclosure of port performance data, as well as with the accountability to internal and external stakeholders working towards the goal of sustainable development.

Change: unify all the concepts to be the same

The whole manuscript

  1. “adaptation of digital transformation towards a sustainable smart port development”

Do you intend to achieve sustainable smart port development or port sustainability performance? What is the endogenous variable anyway?

All the terms have been modified to be (achieve sustainable smart ports performance)

The research aims to investigate to what extent the Egyptian ports could apply smart practices and employ technology to achieve and improve port sustainable performance from the smart perspective that’s why the author adapt the index mentioned in comment number 5 that determine the dependent and independent variables by considering the five domains which are (operation, energy, safety& security, environment, and human resources) in relation to port sustainable performance and its levels (economic, social and environment), consequently, the five domains of smart port practices are considered as independent pillars, while the sustainable port performance is considered as a dependent pillar and technology is counted to be a mediating role between smart practices and port sustainable performance based on previous studies that proved that before

Change: We agree with the reviewer that this seems to be misleading so a paragraph and a figure have been added to identify the research variables and describe the scheme of the research framework on page 13

The whole manuscript

  1. If “Based on the above discussion, it has been proved from previous studies that the technology variable employs and effects the relationship between smart port practices and sustainable performance, which enhance and improve the overall port performance. Different approaches, models and measures have been proposed by many researchers to employ technology to enhance sustainable performance which will be illustrated in next subsection”, what is the GAP referred by authors that this study is filling?

The subsection cited the similar papers and the author identified the research gap that although different approaches, models, and measures have been proposed by many researchers to employ technology to enhance sustainable performance it has been found that there is still a need for an integrated performance measurement system capturing the relationship between smart port practices, and different aspects of port sustainable performance while employing technology, in additions, test to what extent this can be applied on one of the developing countries which is Egypt

Change: Accordingly the research gap was identified and modified in the last paragraph of section 2.2 from citing the previous work in the same area on page 8

Section: 2.2

Page: 7

9.      “that there is still a need for an integrated performance measurement system capturing the relationship between smart port, and sustainability while employing technology” Is technology an exogenous explanatory variable or a moderating variable in the relationship under study? Clarify the role of this variable. Do we have smart port management technologies and practices as causal variables in the relationship or is one moderator or mediator?

From the previous study, it is concluded that the technology variable mediates the relationship between the smart port index and sustainable performance. We agree with the reviewer that not mentioning this seems to be misleading and have amended this section

Change: 2 paragraphs have been added to clarify the variable and the last paragraph in this section has been modified

Also, a paragraph and a figure have been added to identify the research variables and describe the scheme of the research framework

Section: 2.1

Page: 6

Section 3.1

Page 13

10.  “this paper will follow the smart port practices proposed in the above mentioned study [7] the drivers to sustainable port performance”

Now technology disappears and management becomes practices. It has to be very well defined and keep the concepts. They are different concepts. Are they management or operational practices, cultural or organizational, technological? To clarify?

We agree with this reviewer that the information would be useful by unifying the inconsistent concepts to be smart port practices, technology employment, and port sustainable performance

In this section the mentioned study that the paper will follow tackle the indices without including technology employment and its influences on the relationship between smart port practices and sustainable performance. And as mentioned in comment number 8 many previous work proved that technology variable employs and affects the relationship between smart port practices and sustainable performance. Consequently, adding technology to the index as a mediator will enhance and improve the overall port sustainable smart performance, this has been clarified on page 13 and the proof of that is identified in Section: 2.1. Page: 6

Overall the manuscript

11.  Sustainable port performance or port sustainability performance? Are completely different concepts. It is one thing to have an overall sustainable long-term performance in the port and quite another to have a good performance in terms of economic, social and environmental sustainability specifically

The term has been unified in the whole manuscript to be (port sustainable performance) as it’s the main concept, it deals with the measurement and disclosure of port performance data, as well as with the accountability to internal and external stakeholders working towards the goal of sustainable development in terms of economic, social and environmental dimensions 

The whole manuscript

12.  "this research is trying to fill this gap while testing, at the same time, the applicability factors instead of theoretical facts. " The facts are theoretical?, the research of theoretical hypotheses can be based on case studies or on large samples, quantitative, qualitative data and variables, etc. please correct.

The research will follow one of the most comprehensive studies dedicated to the majority of the previous literature, while this study tests the variables theoretically only, 2 stages are needed in order to test the applicability factors

·         The first stage, is to test its suitability to be adapted through conducting interviews and a focus group to investigate the level of readiness and adaptability of the five dimensions that were already measured, analyzed, and mentioned previously in the guideline index, and to test if that could be implemented in the Egypt case or not, and investigate the challenges and obstacles faced the practical adaptation (which identified in this paper) once it's recognized

·         The following stage will be to investigate the impact of adaptation in terms of its impact on the performance and this step will be investigated in another paper (which will be my next step in this project through conducting a survey), as well as that was suggested in the further research as an open issue for future researcher

The research gap highlighted in Section: 2.2, page: 11

Further research section 7, page: 28

13.  RQ3 - please format the point

Thanks for highlighting this typo mistake which has been corrected

Section: 3

Page: 11

14.  Materials and Methods – I don't see any question that leads to confirm the hypothesis that smart ports management and technology can affect port sustainability performance. Were there more questions?

In our study, we highlighted that there is no need to prove that technology is a mediating variable and that technology employment affects the relationship between smart port and sustainable performance as it is already proved before in many previous studies so we will not need statistical analysis consequently, this part has been highlighted in the literature review section with mentioning many previous studies that confirmed the relation.

Also, our study does not include statistical analysis that collects and analyzes large amounts of data. A qualitative research approach has been selected that represents narrative research (a hypothesis is used in the form of a clear statement concerning the problem to be investigated not testing); by conducting interviews and verifying the results with a focus group

15.  "identify the main challenges and obstacles for adaptation" – It is not the aim of the study.

The research aim clarified that the research should come up with the challenges and procedures and how to deal with them in order to adapt the smart port practices linked to sustainable performance as it aims at investigating to what extent the Egyptian ports could apply the smart practices and employ technology to achieve and improve sustainable, performance

This could be achieved through assessing, in the beginning, the current performance and situation of the Egyptian port performance and which were generated from the use of a variety of secondary data concerning Egypt from books, online references, periodicals, and specialized journals in sustainability and maritime transport, work process documents and handouts, plus other public documents that are mainly published in literature and articles that’s why we added this section to the literature review part. Then investigating the level of readiness and adaptability to smart port practices and technology employment, as well as the obstacles and barriers concerning adaption in the Egyptian ports, then suggest solutions to those barriers

And at the end, the author summarized all the barriers and the suggestions in table 2 page 25

This is highlighted in:

·         Abstract page 1

·         Introduction section 1 page 4

·         research gap identification section 2.2, page 11

·         Questions section page section 3, page 11

·         Finding and analysis section 4.1 page 19 &20

·         recommendations of interviews section 5.2 page 25

16.  Authors must define well which study variables are explanatory, moderating, explained and the causal model to be proven. Make a table with the variables and which authors used them previously.

Thanks to the reviewer for pointing this out; a paragraph and a figure have been added to identify the research variables and describe the scheme of the research framework

Section: 3.1

Page: 13

  1. What are smart port management and technology? What are its sub-variables based on the literature?

Subsection 2.1 cited and proposed different approaches, models, and measures by many researchers to employ technology, while subsection 2.2 tackled smart port variables mentioned by many previous studies. In addition, one of the most comprehensive studies dedicated to the majority of the previous literature, tackling and measuring the variables used in detail has been mentioned in detail and adopted as a guideline for our study.

Section: 2.2

Page: 7

18.  What is the concept of port sustainability performance, what are its variables?

The mentioned index that was taken as a guideline for this study stated that there are five main groups that govern the smart port which are the environmental group, the operations group, safety and security, the human factor and the energy group, and ensured that by adopting smart port index, a safe, economical, convenient, green, and sustainable improvement will be enabled; ensuring that it meets the needs of current and future generations in terms of economic, social and environmental aspects

Change: a paragraph has been added before the framework of the study

Also in the literature review, the author highlighted the 3 dimensions of port sustainable performance cited from previous studies

Section: 3.1

Page: 13

Section: 2.1

Page: 5

19.  How do the first ones influence the economic, social and environmental aspects?

The author investigated the level of readiness and adaptability through the five dimensions that have already been measured and analyzed before in the mentioned indices found in the literature, it is observed that an integrated smart port index will enhance the smartness of port processes and operations and affect sustainable performance and its levels (economic, social and environment) since it helps policymakers, directors, and administrators to maintain the flows of their processes in an efficient way by decreasing the consumption emissions, pollution, cost, and time, ensuring risk assessment, productivity, speed, safety, security, flexibility, and knowledge in decision-making processes. Based on these results, and after adding the technology factor as a mediating variable based on the previous study, our study tests to what extent those dimensions could be adapted in the Egyptian context through conducting interviews with experts and verifying the results with a focus group

  1. “Egyptian current performance and current projects” –

Thanks to the reviewer for pointing this out, the author modified this sentence to be (evaluate the Egyptian current performance and situation)

Section: 5

Page: 22

  1. sustainable performance or port sustainability performance?

The concept in the whole manuscript has been unified to be (port sustainable performance)

  1. Why is SWOT analysis relevant for the study goals? Should this part be in the study? Why? No add value to prove the hypothesis. The hypotheses have to be revealed clearly and the conclusion for each one must be shoed, including for each variable.

 To achieve the aim of this study the research should come up with the challenges and procedures and how to deal with them to adapt the smart port practices linked to sustainable performance. The conventional SWOT analysis is based on qualitative analysis. It facilitates an understanding of the strengths and weaknesses. It encourages the development of strategic thinking. It enables senior managers to focus on strengths and build opportunities and determines barriers and obstacles. consequently, (SWOT) analysis has been performed to see the ability of the Egyptian ports in adapting to the novel index by evaluating their current performance and situation as well as their level of readiness to know their possibilities, and opportunities, determine the weaknesses points and external threats in order to draw findings and suggest procedures and solution concerning adaptation

Reviewer 3 Report (New Reviewer)

The study investigates to what extent the developing countries could apply the smart practices and employ technology to achieve and improve sustainable performance of port by taking the ports system in Egypt as an example. By the narrative method of study and based on the interviews with 10 experts in the concerning filed, lots of implications have been illustrated. Basically, it is a good work. However, the biggest drawback is that there are too many redundant contents in the manuscript and lots of English errors. For example:

1.      The contents in Line 11- Line 19 should be compressed substantially.

2.      The contents in Line 55- Line 65 (concerning “The pandemic of COVID-19”) have nothing to do with this study.

3.      In the “Literature Review”, there are lots of concept or idea explanations, such as contents in Line 160- Line 166; Line 167- Line 179. Especially the “2.3. Egyptian ports” should not be included in the  “Literature Review”. It should be in part of “1. Introduction” or3. Materials and Methods”.

4. The tile of “Investigating smart port management and technology  employment influences on ***”might be “Investigating the influences of smart port management and technology employment on ***”

Author Response

Response to Reviewers

Cover Letter

Journal of Sustainability

Dear Editor,

Subject: Submission of revised paper “Investigating smart port management and technology employment influences on port sustainability performance: An empirical study on Egypt” Manuscript ID: sustainability- 1820430

Thank you for giving us the opportunity to submit a revised draft of the manuscript “Investigating smart port management and technology employment influences on port sustainability performance: An empirical study on Egypt” for publication in the Journal of sustainability. We appreciate you and the reviewers for your precious time in reviewing our paper and are grateful for the insightful comments on and valuable improvements to our paper. Your valuable and insightful comments led to possible improvements in the current version. The authors have carefully considered the comments and suggestions made by the reviewers and tried our best to address every one of them.

We hope the manuscript after careful revisions meet your high standards. The authors welcome further constructive comments if any. Below we provide the point-by-point responses to the reviewers’ comments and concerns. All modifications in the manuscript have been highlighted in red. All page numbers refer to the revised manuscript file with tracked changes.

 Sincerely,

Name: Alaa Othman, PhD. alaa.abomousa@gmail.com

The University of Maribor, Faculty of Logistics

Reviewer 1

[General Comment]: The study investigates to what extent the developing countries could apply the smart practices and employ technology to achieve and improve sustainable performance of port by taking the ports system in Egypt as an example. By the narrative method of study and based on the interviews with 10 experts in the concerning filed, lots of implications have been illustrated. Basically, it is a good work.

Response: Thank you for the comments and suggestions on our manuscript entitled. We appreciate the suggested modifications and have revised the manuscript accordingly. We hope the manuscript has been improved accordingly.

However, the biggest drawback is that there are too many redundant contents in the manuscript and lots of English errors. For example:

All the manuscript has been revised in terms of English and grammar, errors, and redundancy

  1. The contents in Line 11- Line 19 should be compressed substantially.

We agree with the reviewer and have amended and modified this part

  1. The contents in Line 55- Line 65 (concerning “The pandemic of COVID-19”) have nothing to do with this study.

Previous studies agreed that the world after the COVID-19 pandemic has realized that there is a critical need for inter-governmental organizations, governments, and industry stakeholders dealing with maritime trade and logistics to come together and accelerate the speed of digitalization so that port communities try to use electronic commerce and data exchange, in observation with all appropriate contractual and regulatory obligations. And this will affect the volume of data, increase the need for mobility in logistics, and the exchange of information which consequently directed to increasing demand for data security and data protection in maritime logistics to stop manipulations of sensitive systems; thus, all players in the maritime supply chain will have to provide the best attainable protection to defend their data against unauthorized access and any kind of abuse by cloud-based user approaches, strong system, access management, and data backup, also the effect of the pandemic on the maritime management, will affect many domains in the performance of the ports and that’s what the interviewees highlighted.

  1. In the “Literature Review”, there are lots of concept or idea explanations, such as contents in Line 160- Line 166; Line 167- Line 179. Especially the “2.3. Egyptian ports” should not be included in the  “Literature Review”. It should be in part of “1. Introduction”or “3. Materials and Methods”.

We agree with the reviewer's point of view many paragraphs have been deleted or modified

  • Line 160- 166 (deleted)
  • Line 167- 179 (modified)
  • 3 “Egyptian ports” the author aims in this study to investigate the challenges and procedures and how to deal with them in order to adapt the smart seaport practices linked to sustainable performance through assessing, in the beginning, the current situation of the Egyptian port's performance and that was generated from the use of a variety of secondary data concerning Egypt from books, online references, and periodicals and specialized journals in sustainability and maritime transport, work process documents and handouts, plus other public documents that are mainly published in literature and articles that’s why we added this section to the literature review part. Then investigating the level of readiness and adaptability to smart port practices and technology employment in the Egyptian ports which have been illustrated in section 4
  1. The tile of “Investigating smart port management and technology employment influences on ***” might be “Investigating the influences of smart port management and technology employment on ***

We agree with the reviewer that this would be better

Change: The title has been updated to the following: Investigating the influences of smart port practices and technology employment on port sustainability performance: The Egypt case

Round 2

Reviewer 2 Report (Previous Reviewer 3)

Authors have improved the paper quality.

Reviewer 3 Report (New Reviewer)

no further commnents

This manuscript is a resubmission of an earlier submission. The following is a list of the peer review reports and author responses from that submission.

Round 1

Reviewer 1 Report

Authors conduct Assessment the Egyptian ports  and their adaptability to smart practices linked to sustainable performance. The idea is good,  but a lot has to be done to have scientific soundness.

ABSTRACT: some statements (content) are repeating, methodology part  is missing.. to be verified and summarized

line106-7:"Section 5 of this paper proposes the  conceptual framework and discusses the obtained findings and results ."

- proposed conceptual framework in the Discussion but research questions are stated in the subchapter 3.3 ? to be elaborated.. what is the real goal of the study... confusing

line 417-19: "In the frame of narrative research, the collected information is concentrated and retold. Building upon this, the research was completed by field research and observations, " 

- retold?! this can be very complicated; either summarized or quoted .

-which observation... be specific and accurate; there are no observations in  the text stated

line 433-35: "The authors of the present study performed expert interviews with 10 stakeholders  that have experimented with this specific subject and experienced it in their previous work."

- experimented on what? this is very ambiguous and unclear

- who are really the "stakeholders"

line 446 : "The interviews fell into two parts: ...." is the same sentence as in line 458

Lines 583-590: "4. The interrelation between sustainable practices and smart practices......................................." ??????

- interrelations is measured and proved by appropriate scientific methods, which are not presented in the paper?!

Lines 594-600: "The pandemic practically trained the world that  sustainable development should be the greatest objective and that normal trading  depending on human resources jeopardizes development and puts people and the planet  at risk, in addition to the building of deeper terminals and logistics areas and accelerating  the pace of digitalization, which will enhance the loading and unloading process, reduce the waiting times of ships, improve the skills of customs employees and enable customers to finish their papers in one place, which will reduce the cost of custom clearance.

-this sentence is unusual long a hard to apprize. To be verified or split

Lines 710-711: "Others stated that in order to adapt sustainable smart port in Egypt, it is required to  have manufacturing certificates and sailing certificates."

- who are the others

- manufacturing certificates and sailing certificates ?? pls. elaborate further this statements

Lines: 721-727:"Others agreed that Construct ships with modern innovation and quality use the  renewable energies system for the ships’ designs, move towards integration intermodal  transport techniques, decrease emissions, spread over the tracking and tracking  technologies, and focus on more automation and integration of data, promote investment  in technology, enhance the environmental and waste management systems, adopt the  corridor management strategies, and develop and implement sustainable energy action  plans, which will facilitate the vision of adopting SSP practices." 

 -Construct ships and innovation...?? pls elaborate and provide source for this statement

Lines760-2: "Based on the previous work, research problem, and gaps, SWOT analysis for the port  is subject to the data collected from the analysis of the above interviews, in addition to the  author's research." 

- what are the data collected?

- what is the authors research ?

line 785-6: "More importantly, the communication and IT sector in Egypt has perceived an  extraordinary growth from 2016 until now, with a yearly average growth of 13.7%."

- source for this statement is missing!

line 839: "5.4 Threads"... to be corrected

Lines 878-9: "Therefore, from the previous study, it is concluded that the technology variable  mediates the relationship between the smart port practices and sustainable performance." 

- based on previous study,  authors drew the conclusion that technology mediate relationship ...? to prove mediation appropriate statistical analysis should be applied. No proof that any was conducted in this study

Lines 880-885: "Considering interviews’ feedback, in addition to the comprehensive study of  technology, the previous steps are to investigate and structure the conceptual frame work.  Figure 5 illustrates the proposed framework for this study, where the dependent variable  is the Sustainable Performance (Environmental, Social and Economic dimensions). Also,  the independent variable is Smart port practices (Operation, Environment, Energy, Safety  and Security and Human factor). In addition, the mediator variable is Technology." 

- introducing the framework at this stage of study is strange. Now you are stating dependant variable and independent variables. Again, appropriate statistical analysis should be applied. No proof that any was conducted in this study. Same issue regarding the mediator variable.  all of them (variables) should be measured and results presented; missing!!

Lines: 889-894: "This study investigated the current situation of the Egyptian port's performance as  one of the developing countries, identified the level of readiness and adaptability to smart  port practices, and linked them to sustainable performance, followed by investigating the  main obstacles and challenges associated with their implementation. The empirical study  of Egyptian seaport showcased that the digitalization and smart port practices can be  successfully adapted."

- port performance must be measured and analyzed to drew some info and  conclusion

- level of readiness and adaptability to smart  port practices - this issue must  be measured and analyzed to drew some info and  conclusion

- to drew conclusions based on 10 respondent opinions (unknown background and experience) is very shaky/ambiguous, at least?!

..................................................

Presented manuscript should in general be shortened,   same sentences and issues issues are repeated.

Numerous statements are missing adequate source.

Using statistical terms should be backed with appropriate statistical analysis.

Similar studies should be cited and  results mentioned/compared  in the manuscript.

Reviewer 2 Report

The topic proposed by the author of the article is undoubtedly interesting and actual, because in the context of changes in global logistics processes, the use of smart technologies will ensure the sustainability and safety of maritime transportation and the transport system as a whole. The content of the article as a whole adequately reflects the formulated problem, and its title correlates with the text. The abstract describes the essence of the article, briefly outlines the state of the problem, research methods, and results. Keywords adequately reflect the content of the article.

In the introduction, the authors provide a brief overview of the state of the problem, indicate the goal and tasks of the study, as well as a description of the article structure. Section 2 reviews the literature on smart ports and sustainability, characterizes Egyptian ports, and studies on the level of digitalization and digital initiatives of commercial ports. Section 3 is devoted to describing the methods that the authors used to conduct the study. Section 4 presents an analysis of the study results based on its objectives. Section 5 is devoted to presenting the conceptual framework and discussing the results obtained. In section 6 - "Conclusions" the authors summarize the results obtained and draw conclusions on the work. Section 7 outlines the limitations of this study as well as possible directions for future research.

The article is relevant to the topic it explores and presents, the article is prepared in accordance with the instructions for the authors. In our opinion, the article is in line with the theme of "increasing the intelligence of maritime logistics" and corresponds in type to the Preliminary Study.

Comment.

1.  The topic is interesting, but the vaguely formulated goal and tasks of the study, as well as the contribution that the authors of the article make with their research to the development of maritime transportation and the sustainability of the seaports functioning.This fact does not allow us to understand how the ideas put forward by the authors can be realized.

2.  The discussion is rather superficial, since the applied SWOT analysis method is not sufficiently described from the point of view of analytics, because the authors did not indicate what the consequences of certain problems and threats will be, and also how advantages and strengths can be used to solve problems. In addition, the authors did not indicate how this might be used for future research.

3.   In general, it is not clear how the review of the literature reflects the scientific or practical content of the available research, including which of the ideas can be applied to the ports of Egypt.

4.   There are comments on the design of the literature list (there are typos) and the quality of the figures (in some cases, the inscriptions are hard to read), it is necessary to improve the clarity of the images.

Reviewer 3 Report

An important research issue

Smart port development (SPD) has been currently regarded as the top priority for every international benchmark port.

It is not easy to understand how the variables can allow classifying the smart port since it includes many operational indicators that must be better explained.

Seem to be more related to sustainable Port. Nothing is seen about big artificial intelligence, data, smart Gate, logistical transparency, physical Internet.

It seems after all we are not dealing with smart ports but only with sustainable ports, so perhaps it is necessary to rethink the all context.

 I would say that we are looking more at an investigation of how Port management influences sustainability policy and development and Port operational performance.

Smart port has been currently regarded as the top priority for every international benchmark port.

Is it possible to build a table with the main variables used and authors?

What is the main hypotheses from authors gap?

Round 2

Reviewer 1 Report

Dear Authors,

your reply was  well received. My first impression after reading this version is that overall looks even worse than the original, the second and third reading confirmed the impression. New version is obviously hastily written.

There are numerous issues related with the manuscript, I will point out the main ones.

Authors Reply on the previous comments:

comment 3: which observation... be specific and accurate; there are no observations in the text stated....this comment is ignored.... which observations ?

also, reply to other comments is impossible to track or answers were not provided.

Most important issues summarized in comments 13, 15 18 , i.e., usage of statistical analysis were simply ignored and removed. I expected that collected "data" will provide some insight into stated... Such behaviour of the authors is Unacceptable.

Furthermore, in the new version following issues are also problematic:

In the literature review, lines 264-279, authors specify something, refering to the source [7], ...and continue till line 275... The study test.... From the stated it is practically impossible to know on which study they refer, [7] or their own ?

subsection 2.3 Empirical study on Egiptian ports.... 3 pages.... what is the purpose of describing Egyptiian ports in such manner? recommendation for summarizing was not observed

lines 435-440 - this is confusing/ unreadable 

Methods, line 499... meetings duration 1 hour; line 514: meetings 30 to 50 min ???  

line 649 - : The relationship between sustainable practices and smart practices.

"All of the interviewees confirmed that there is a positive relationship between sustainability and smart practices; they agreed that innovating in sustainability and technology will positively affect the entire port’s ...."

- To prove relationship (positive or negative) application of appropriate analysis is required. same comment as the previous time.

line 767-770: "Other experts stated that in order to adapt sustainable smart port in Egypt, it is required to have marine certificates which allow worker to be capable to work onboard ships according to the provisions of the International Maritime Organization (IMO) and the International Convention on Standards of Training."

- STCW is for seafarers (for serving onboard) mandatory .how the expert relate this issue with sustainable port ?! 

In short, presented manuscript (original and revised version) still require a lot of work to have some scientific soundness.

Author Response

Response to Reviewers

Cover Letter

Journal of Sustainability

Dear Editor,

Subject: Submission of revised paper “Investigating smart port management and technology employment influences on port sustainability performance: An empirical study on Egypt” Manuscript ID: sustainability- 1820430

Thank you for giving us the opportunity to submit a revised draft of the manuscript “Investigating smart port management and technology employment influences on port sustainability performance: An empirical study on Egypt” for publication in the Journal of sustainability. We appreciate you and the reviewers for your precious time in reviewing our paper and are grateful for the insightful comments on and valuable improvements to our paper. As a researcher, it's always my pleasure to accept comments from reviewers which make my piece of work better and this is how the researcher always learns. The authors have carefully considered the comments and suggestions made by the reviewers and tried our best to address every one of them.

We hope the manuscript after careful revisions meet your high standards. The authors welcome further constructive comments if any. Below we provide the point-by-point responses to the reviewers’ comments and concerns. All modifications in the manuscript have been highlighted in red. All page numbers refer to the revised manuscript file with tracked changes.

 Sincerely,

Name Alaa Othman, Ph.D. alaa.abomousa@gmail.com

The University of Maribor, Faculty of Logistics

Reviewer 1

[General Comment]: your reply was well received. My first impression after reading this version is that overall looks even worse than the original, the second and third reading confirmed the impression. New version is obviously hastily written.

There are numerous issues related with the manuscript, I will point out the main ones.

Response: Thank you for your comments. I would like to clarify that I am as a researcher always respect all comments received from reviewers and open to any changes that enhance the research in a better way, I am not sure why it perceived that I ignored some comments although I already fulfilled, however, I tried to do my best again to clarify the previous comment and how it’s done and also highlighting the new comments

While we appreciate the reviewer’s feedback, we think this study makes a valuable contribution to the field because a smart port is a broad concept that contains several aspects of port activities. Although many previous studies discuss smart ports requirements there is limited availability of literature and research from academic researchers that captured different elements of smart port and technology employment and show their influences on sustainable performance, in addition, test to what extent this can be applied on ports, particularly in developing countries, also there is a lack of literature that addressed the current situation of the Egyptian ports’ performance and identified the main challenges for smart practices’ adaptation from different perspectives; therefore our research will fill this gap by investigating to what extent the developing countries could apply the smart practices and employ technology to achieve and improve sustainable performance, particularly in Egypt.

We appreciate the suggested modifications and have revised the manuscript accordingly. We hope the manuscript has been improved accordingly.

The detailed point-by-point responses to the reviewers’ comments are presented in detail with referring to the sections, pages and lines changed in the (tracked manuscript) as follows:

Reviewer 2 Comments

Author Response

Location of Response in Revised Manuscript

Old Comments

1.      Authors Reply on the previous comments:

comment 3: which observation... be specific and accurate; there are no observations in the text stated.... this comment is ignored.... which observations?

This comment already matched in the version submitted, the author agreed with the reviewer’s point of view observation is not the case in our study, so in the response, we wrote that (the paragraph has been restructured to point out the methodology used), and we mentioned the page and specifically the parts that have been deleted and replaced with the appropriate method in the (previous tracked version) by writing comment 3, Reviewer: 1

The change that the author already did is that: The word “field research and observation” have been deleted and replaced with “conducting semi-structured interviews to converse with experts and collect elicit data”

However, I will highlight this again and write a comment on the tracked manuscript

Section: 3.1

Page: 13

Line: 591, 592

2.      also, reply to other comments is impossible to track or answers were not provided.

The author respects the reviewer's comment, we have rechecked all previous comments, the responses, and the way of tracking them and we will do our best to clarify all the comments received and illustrate them in a better way by writing the page, section and lines modified also highlighting the changes by writing comments (in red) in the tracked manuscript

3.      Most important issues summarized in comments 13, 15 18, i.e., usage of statistical analysis were simply ignored and removed. I expected that collected "data" will provide some insight into stated... Such behavior of the authors is Unacceptable.

a.       Comment 13: we agreed before with the reviewer’s point of view that proving mediation needs appropriate statistical analysis.

In our study, we highlighted that there is no need to prove that technology is a mediating variable and that technology employment affects the relationship between smart ports and sustainable performance as it is already proved before in many previous studies so we will not need statistical analysis consequently, this part has not been deleted but moved after modifying to the literature review section with mentioning many previous studies that confirmed the relation.

b.      Comment 15: This comment has 3 main points, in the previous response we have classified them as a, b and c, however, fine-tune has been made to highlight the data collected used

We did not ignore the comments by deleting them, we agreed before with the reviewer’s comment that the flow of the methodology part seems to be misleading and unclear, so we have revised the whole manuscript, which definitely reflects on the methodology section by deleting and adding paragraphs, summarized in the following points:

·         A qualitative research approach has been selected that represents narrative research; interviews are the most commonly used data collection method in qualitative research and the foremost method in narrative inquiry in particular, as we understand that interviewees will answer the questions based on their narrative schema that reflects their knowledge and experiences, so our study does not include statistical analysis that collects and analyze large amounts of data

·         The research investigates the level of readiness and adaptability through the five dimensions that were already measured and analyzed before in previous studies mentioned specifically on page number 6, by explaining the index dimensions, findings, limitations, and further research. Our study adapts those dimensions to the Egyptian context by conducting interviews with experts mentioned in table 1

·         The study measures the influences on port performance by asking questions that measure the impacts on the economic, social, and environmental dimensions

·         (SWOT) analysis has been performed to see the ability of the Egyptian ports in adapting the novel index

c.       Comment 18: the reviewer commented that using statistical terms should be backed with appropriate statistical analysis; the study does not use any statistical term referring to points (a and b) mentioned above, and the whole manuscript has been modified based on the points stated earlier which highlight that the research use qualitative not quantitative approach. 

A

Section: 2.1

Page: 5

Line: 234 to 281

B

Section: 2.2

Page: 7

Section: 3.1

Page: 13

New Comments

4.      Furthermore, in the new version following issues are also problematic:

In the literature review, lines 264-279, authors specify something, referring to the source [7], ...and continue till line 275... The study test.... From the stated it is practically impossible to know on which study they refer, [7] or their own?

We agree with this reviewer that this seems to be misleading, the study will adopt the novel index referred to in source number 7  which is mentioned in detail in the literature section on page number 6, however, this paragraph has been modified to differentiate between the study adopted and our research

Section: 3.1

Page: 13

Line: 599 to 607

5.      subsection 2.3 Empirical study on Egyptian ports.... 3 pages.... what is the purpose of describing Egyptian ports in such manner? recommendation for summarizing was not observed

Many paragraphs and 2 figures have been removed in the (previous tracked) manuscript and those lines have been removed

·         line 355 to 359

·         line 330 to 336

·         line 415 to 436

·         line 472 to 490

·         line 498 to 502

·         line 517 to 530

however, to avoid redundancy and shorten the manuscript, other paragraphs have been removed in the empirical study section, kindly check the following lines in the (recent tracked) version

·         line 336 to 346, p: 7

·         line 451 to 471, p:10

·         Line 376 to 379, p: 8

·         Line 389 to 392, p: 8

·         Line 504 to 571, p: 11

Section: 2.2

6.      Lines 710-711: "Others stated that in order to adapt sustainable smart port in Egypt, it is required to have manufacturing certificates and sailing certificates."

A.    who are the others

B.     manufacturing certificates and sailing certificates ?? pls. elaborate further this statements

A.    Other experts that are mentioned in table 1 in section 3 elaborates the stakeholders (experts) with their year of experience, type of organizations, and positions

B.     Statements have been added with examples for the training needed instead of manufacturing and sailing certificates,

A

Section: 3.2

Page: 15

B

Section: 4

Page: 20

line 949 to 953

7.      lines 435-440 - this is confusing/ unreadable

The author agrees with the reviewer’s comment, we have checked the lines in the (clear version) and modified them accordingly in the tracked version

Section:3

Page: 12

 Line: 565 to 570

8.      Methods, line 499... meetings duration 1 hour; line 514: meetings 30 to 50 min??? 

We agree that the meetings time should be from 30 to 50 minutes but some meetings were conducted on zoom which took longer due to connectivity issues, however, we will modify this part

Section: 3.2

Page: 15

Line: 660

9.      line 649 - : The relationship between sustainable practices and smart practices.

The sub-title has been modified

Section: 4.4

Page: 18

10.  "All of the interviewees confirmed that there is a positive relationship between sustainability and smart practices; they agreed that innovating in sustainability and technology will positively affect the entire port’s ...."

- To prove relationship (positive or negative) application of appropriate analysis is required. same comment as the previous time.

We thank the reviewer for pointing out this, the positive relationship between sustainability and smart practices has been proved before in the literature and mentioned in the manuscript on page 7 line 304

However, the author understands that mentioning this again without analysis may mislead the reader, consequently, the way of writing has been modified to be (All of the interviewees confirmed that smart practices influence sustainable performance)

Section: 4.4

Page: 18

Line: 818 and 837

11.  line 767-770: "Other experts stated that in order to adapt sustainable smart port in Egypt, it is required to have marine certificates which allow worker to be capable to work onboard ships according to the provisions of the International Maritime Organization (IMO) and the International Convention on Standards of Training."

- STCW is for seafarers (for serving onboard) mandatory. how the expert relates this issue with sustainable port?!

The authors are grateful to the reviewer for pointing out the mistake. All the recorded interviews relating to this part have been revised, and accordingly, this part has been modified in the tracked manuscript.

The author summarized all types of certificates stated by the interviewees by writing (Marine certificate) which now has been removed, and substituted by “energy management certificates or arrangements according to any standard (ISO 50001, etc), number of safety and security arrangements and certificates, certificates in maritime environmental management, operator of lifting equipment training, cargo coordinator training, planner training, and port worker training program in safety and health”

Section: 4

Page: 20

Line: 948 to 952

12.  In short, presented manuscript (original and revised version) still require a lot of work to have some scientific soundness

We appreciate the suggested modifications and have revised the manuscript accordingly. We hope the manuscript has been improved accordingly.

The whole manuscript

Reviewer 3 Report

Dear authors

The paper could benefit from a variable list with authors that could organize the results.  

Author Response

Thanks to the reviewer for his comment, The results of the study are based on a qualitative research approach which represents narrative research; interviews are the most commonly used data collection method in qualitative research and the main method in narrative inquiry in particular, as we understand that interviewees (listed in table 1) will answer the questions based on their narrative schema that reflects their knowledge and experiences and the data presented are extracted from them to evaluate their level of readiness and adaptability of smart practices and technology in all port activities and identify the main challenges and obstacles for adaptation.